# Mechanism of membrane-curvature generation by ER-tubule shaping proteins

Ning Wang[1], Lindsay D. Clark [1], Yuan Gao[1], Michael M. Kozlov [2], Tom Shemesh[3] & Tom A. Rapoport [1✉]

The endoplasmic reticulum (ER) network consists of tubules with high membrane curvature in cross-section, generated by the reticulons and REEPs. These proteins have two pairs of trans-membrane (TM) segments, followed by an amphipathic helix (APH), but how they induce curvature is poorly understood. Here, we show that REEPs form homodimers by interaction within the membrane. When overexpressed or reconstituted at high concentrations with phospholipids, REEPs cause extreme curvature through their TMs, generating lipoprotein particles instead of vesicles. The APH facilitates curvature generation, as its mutation prevents ER network formation of reconstituted proteoliposomes, and synthetic L- or D-amino acid peptides abolish ER network formation in *Xenopus* egg extracts. In *Schizosaccharomyces japonicus*, the APH is required for reticulon's exclusive ER-tubule localization and restricted mobility. Thus, the TMs and APH cooperate to generate high membrane curvature. We propose that the formation of splayed REEP/reticulon dimers is responsible for ER tubule formation.

[1] Howard Hughes Medical Institute and Department of Cell Biology, Harvard Medical School, Boston, MA 02115, USA. [2] Department of Physiology and Pharmacology, Sackler Faculty of Medicine, Tel Aviv University, Tel Aviv 69978, Israel. [3] Faculty of Biology, Technion-Israel Institute of Technology, Haifa 32000, Israel. ✉email: tom_rapoport@hms.harvard.edu

All cellular organelles have characteristic shapes, but how their morphology is generated is largely unknown. The endoplasmic reticulum (ER) is a particularly intriguing organelle, as it consists of morphologically distinct membrane domains, including a characteristic network of interconnected tubules[1,2]. Interspersed in the network are sheets containing closely apposed membranes[3]. ER tubules have a diameter of 30–60 nm[3,4] and thus have high membrane curvature in cross-section. A similar high curvature is seen at sheet edges[3]. The tubules themselves are shaped by two evolutionarily conserved protein families, the reticulons (Rtns) and REEPs (Yop1 in yeast)[5,6]. In the absence of these proteins, tubules convert into sheets, but the tubular network formation can be restored when at least one of these proteins is expressed at a sufficient level[3,5,6]. When purified Rtn or REEP proteins are reconstituted into liposomes, they form short, narrow tubules[7]. Finally, these proteins localize exclusively to tubules and sheet edges, the high-curvature regions of the ER[3,5,6].

Connecting tubules into a network requires membrane fusion, which is mediated by membrane-anchored GTPases, the Atlastins (ATLs) in metazoans and Sey1, and related proteins in yeast and plants[8–10]. Proteoliposomes containing a purified fusion GTPase and a member of the reticulon/REEP families form a tubular membrane network in the presence of GTP, indicating that these are the minimal components to generate an ER network[11]. The ATLs also seem to be required for the formation of the tubules themselves, as depletion or inactivation of ATL causes the breakup of tubules into small vesicles or membrane fragments[12].

The high membrane curvature of tubules and sheet edges is stabilized by the Rtns and REEPs by an unknown mechanism. All of these proteins contain pairs of closely spaced transmembrane (TM) segments followed by an amphipathic helix[6,13,14] (APH; Fig. 1a and Supplementary Fig. 1a). It has been proposed that Rtns and REEPs generate high membrane curvature by hydrophobic insertion (wedging) and scaffolding[6,10]. Wedging would involve the shallow insertion of the APH into the cytoplasmic leaflet of the lipid bilayer, which would displace the polar head groups of the surrounding phospholipid molecules and thus produce local membrane bending[6,13–16]. Scaffolding would involve the formation of intrinsically curved arc-shaped protein oligomers on the membrane surface, which would mold the lipid bilayer into tubules. However, whether these mechanisms are responsible for the membrane-shaping activity of the Rtns and REEPs remains unclear. In this study, we use in vitro and in vivo experiments to show that the TMs and APH cooperate to generate high membrane curvature and to derive a model for ER-tubule formation.

## Results

### REEP monomers form tightly interacting dimers within the membrane.

The hydrophobic insertion mechanism of curvature generation can occur with protein monomers and does not require their oligomerization. By contrast, the formation of protein oligomers is essential for the scaffolding mechanism. To understand the mechanism by which REEPs and Rtns generate high membrane curvature, it is therefore important to determine their oligomeric states. As a model protein, we chose the REEP protein Yop1 from the fission yeast *Schizosaccharomyces japonicus* (referred to as Yop1), because this organism was used for in vivo experiments discussed in a later section. Yop1 was expressed in *E. coli* with a C-terminal streptavidin-binding peptide (SBP) tag and purified from the membrane fraction following solubilization in the detergent dodecylmaltoside (DDM). After affinity-chromatography on streptavidin beads and size-exclusion chromatography (SEC), the protein migrated as a single band of

the expected size in Coomassie-blue stained SDS gels (Fig. 1b). It formed a dimer, as demonstrated by SEC combined with multi-angle light scattering (MALS) (Fig. 1c). Similar results were obtained with *Xenopus laevis* REEP5, purified in the same way (Fig. 1b, c).

Dimer formation of Yop1 was confirmed with pull-down experiments. SBP-tagged full-length Yop1 (Yop1-SBP) or a mutant lacking the APH (Yop1ΔAPH-SBP) were co-expressed in *E. coli* with similar versions of His-tagged Yop1 (Yop1-His10 or Yop1ΔAPH-His10). DDM-solubilized membrane fractions were incubated with streptavidin beads, and the bound material was analyzed by SDS-PAGE and Coomassie-blue staining. The results show that the SBP-tagged protein pulled down the His-tagged proteins, regardless of the presence of the APH (Fig. 1d). These results also indicate that dimer formation does not require the APH.

To further confirm dimerization, we inserted a folded protein, a variant of cytochrome b562 (BRIL)[17], into the C-terminal segment of Yop1 (Supplementary Fig. 1b). The purified fusion protein was incubated with Fab fragments of BRIL antibodies[18], and the sample was analyzed by negative-stain electron microscopy (EM) (Supplementary Fig. 1c–e). Most particles showed two dots corresponding to Fab fragments (Supplementary Fig. 1e), consistent with dimerization of the fusion protein. A mutant with a charge introduced into the hydrophobic side of the APH, Yop1 (I145R)-BRIL, also bound two Fab fragments (Supplementary Fig. 1e), indicating that the amphipathic nature of the helix is not required for dimerization.

Having shown Yop1 dimerization in detergent, we next tested whether dimer formation occurs in intact membranes. Photo-reactive 4-benzoyl-L-phenylalanine (Bpa) residues were incorporated into different TMs of Yop1 by suppression of amber codons[19,20]. The purified proteins were irradiated with UV light either in DDM or after reconstitution into liposomes. Cross-linked products were analyzed by SDS-PAGE and Coomassie-blue staining (Fig. 2a). Strong cross-linking was observed when Yop1 was reconstituted into membranes, and the probe was incorporated at F65 into TM2 (Yop1F65Bpa); >75% of the monomers were converted into dimers. Probes in other TMs showed much less cross-linking (Fig. 2a). In detergent, all positions cross-linked only weakly, including F65 (Fig. 2a). We also incorporated photoreactive probes at many positions of the APH, as well as positions preceding or following it, but none gave strong cross-links (Supplementary Fig. 2a, b), supporting the conclusion that the APH is not involved in dimerization. Incorporating Bpa into the N-terminal segment or the cytoplasmic loop between TMs 2 and 3 did not give strong cross-links either (Supplementary Fig. 2b). To confirm that cross-linking occurs between monomers in the dimer, rather than between dimers, we irradiated Yop1F65Bpa in liposomes and then solubilized the membranes in DDM and performed SEC. The cross-linked product eluted at the position of the dimer, rather than at a higher molecular weight (Supplementary Fig. 2c). Taken together, these results show that the REEP protein forms stable dimers, rather than monomers or higher oligomers, as previously assumed[6,21]. The high cross-linking yields with a probe at F65 suggests that TM2 of two monomers tightly interact with one another inside the membrane. Further support for this assumption comes from experiments in which we placed the photoreactive probes at other positions in TM2 (Supplementary Fig. 2d). The cross-linking yields were lower but showed maxima at positions expected to be on one side of a helix formed by TM2 (positions 61, 65, and 69). These results suggest that dimerization is mediated by TM2. This TM is indeed special among the four TMs, as it contains well-conserved amino acids, including a Pro and several hydrophilic residues (Supplementary Fig. 2e).

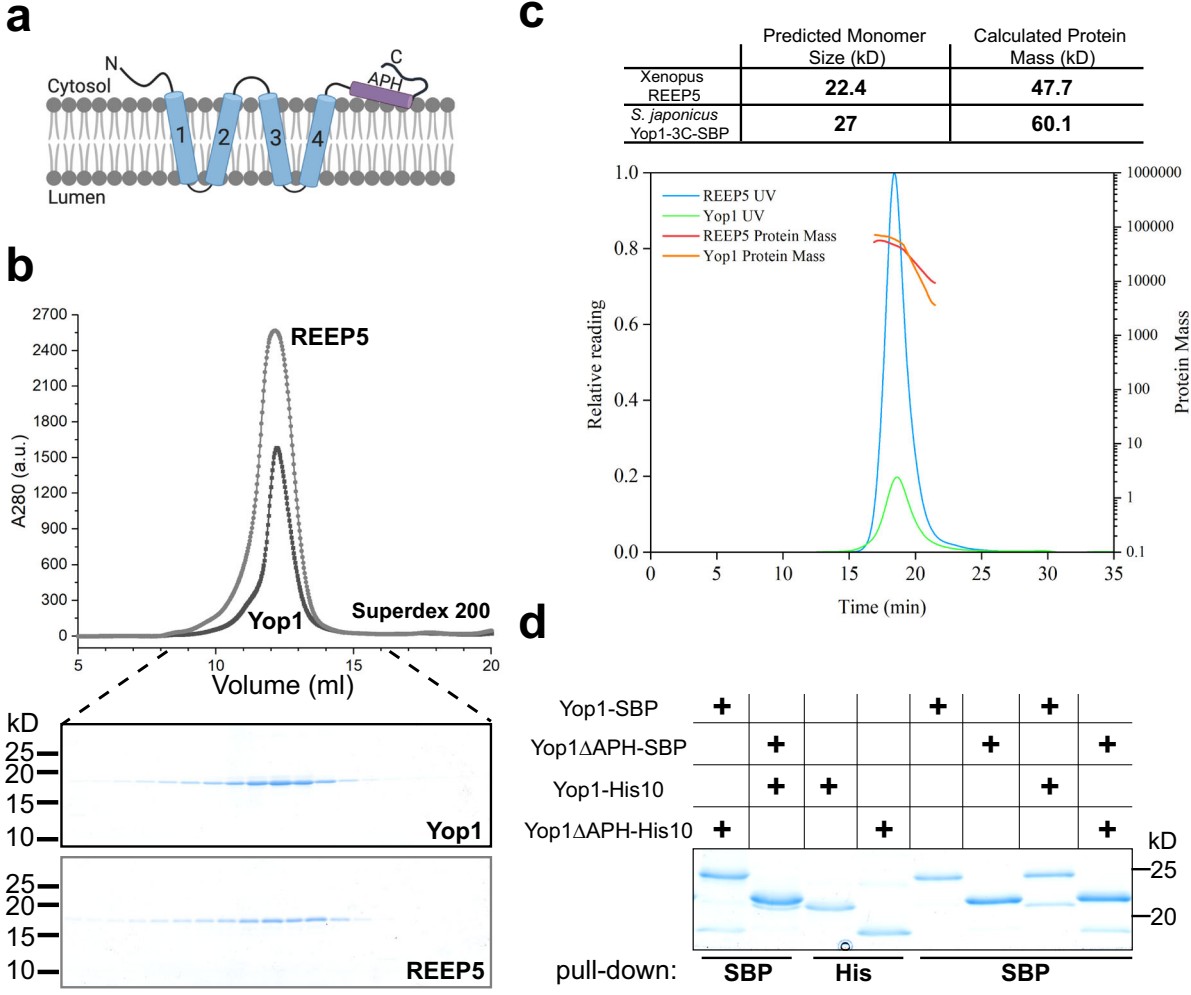

**Fig. 1 REEPs form dimers. a** Proposed membrane topology of the REEPs and Rtns. The four hydrophobic transmembrane segments (TMs) are numbered and shown in blue; the predicted amphipathic helix (APH) is shown in purple. **b** SEC of *S. japonicus* Yop1 and *Xenopus laevis* REEP5 purified in the detergent DDM. The experiment was performed four times. The top panel shows the absorbance at 280 nm. Fractions of the eluate were analyzed by SDS-PAGE, and stained with Coomassie blue (lower panels). **c** SEC-MALS analysis of purified REEP5 and Yop1-3C-SBP, analyzed in DDM. Absorbance at 280 nm (UV) was normalized relative to the maximum reading of REEP5 (left axis). The calculated protein mass is shown across the peaks (right axis). The table on top shows the calculated and predicted monomer masses. **d** SBP-tagged full-length Yop1 or Yop1 lacking the APH (Yop1ΔAPH) was co-expressed with His-tagged full-length Yop1 or Yop1ΔAPH. DDM-solubilized membrane fractions were incubated with streptavidin beads and the bound material analyzed by SDS-PAGE and Coomassie-blue staining. As a control, SBP- and His-tagged Yop1 were individually expressed and purified. The experiment was performed three times.

Deletion of the APH or introduction of charged residues into the hydrophobic face of the helix reduced cross-linking of Yop1 with the probe at F65 (Fig. 2b; quantification see Fig. 2c). Thus, although the APH is not essential for dimerization of Yop1, it contributes to the interaction between the monomers in the membrane.

To test whether dimer formation is a general property of REEPs, we introduced a photoreactive probe at a position equivalent to F65 into *Xenopus laevis* REEP5, and into REEPs from two thermophilic fungi (*Thielavia terrestris* REEP5, and *Thermothelomyces thermophila* REEP5; Fig. 2d). In each case, irradiation of the purified, reconstituted protein gave dimer cross-links, although not with the same efficiency as with Yop1. We conclude that the ground state of REEPs is a dimer in which monomers tightly interact within the membrane, an interaction facilitated by the APH. The reduced cross-linking efficiency in detergent suggests that dimers have a strained conformation in the membrane, which is relaxed upon solubilization.

**Reconstituted REEP generates lipoprotein particles with extreme curvature.** To investigate the membrane-shaping activity of REEPs, we inserted Yop1 into preformed liposomes (reconstitution by directed insertion[22,23]) and analyzed the resulting membrane structures by negative-stain EM. The generated structures were heterogeneous, likely because they contained Yop1 at different densities. The structures consisted of narrow short tubules (diameter 12–15 nm; length up to 600 nm), small spherical particles with a diameter of <12 nm, vesicles with a diameter between 12 and 30 nm, and vesicles with a diameter >30 nm (Fig. 3a; quantifications in Fig. 3b). The small particles (diameter <12 nm) were observed previously with *S. cerevisiae* Yop1 or Rtn1, but were believed to be vesicles[7]. However, some of these particles have diameters smaller than 8 nm, which makes it unlikely that they consist of a lipid bilayer surrounding an aqueous lumen (a bilayer has a thickness of ~4 nm). Rather, these structures appear to be lipoprotein particles (LPPs), i.e., protein/lipid micelles. Given their small diameter, the curvature of these structures is extremely high. When Yop1 lacking

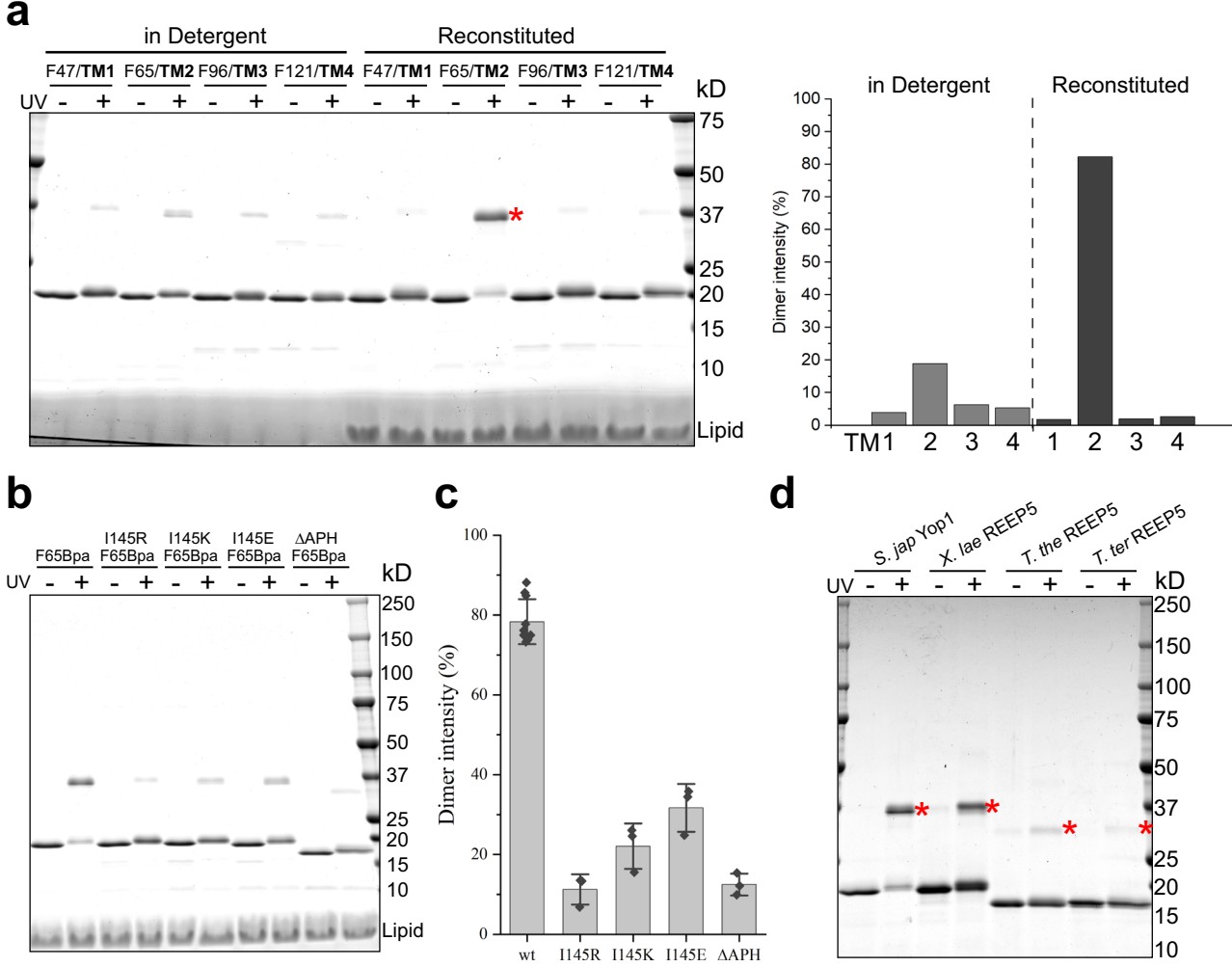

**Fig. 2 REEP/Yop1 monomers interact in membranes through TM2. a** Photoreactive Bpa probes were incorporated into Yop1 at the indicated positions of different TMs. The purified protein was irradiated with UV light either in DDM or after reconstitution in liposomes containing *S. cerevisiae* lipids at a 1:200 molar ratio of protein to lipid. The samples were analyzed by SDS-PAGE and Coomassie-blue staining. The asterisk indicates the position of the cross-linked Yop1F65Bpa dimer. The right panel shows quantification of the intensity of dimer cross-links relative to total protein. **b** As in **a**, but with mutants in the APH reconstituted into proteoliposomes. **c** Quantification of experiments as carried out in **b**. Data are presented as mean ± the standard deviation (SD) from $n = 10$ (wt) and $n = 3$ (I145R, I145K, I145E, ΔAPH) independent experiments. **d** Dimer formation was tested with purified REEPs from *Schizosaccharomyces japonicus* (*S. jap*), *Xenopus laevis* (*X. lae*), *Thermothelomyces thermophila* (*T. the*) and *Thielavia terrestris* (*T. ter*). In each case, a Bpa probe was incorporated at the position of the conserved Phe residue in TM2. The samples were analyzed as in **a**. The asterisks show the positions of cross-linked dimers.

the APH was reconstituted with phospholipids, tubules were almost absent, as previously reported[13], and larger vesicles became more abundant (Fig. 3a, b). The number of LPPs decreased moderately (Fig. 3b). This suggests that LPPs were formed by domains of high Yop1 concentration, whereas domains of low Yop1 concentration required the APH to produce the curvature of tubules and/or small vesicles. When reconstituted wild-type Yop1 was subjected to flotation in a Nycodenz gradient (Fig. 3c), tubules and vesicles were found in the top fractions (Fig. 3d; F2), whereas the LPPs stayed near the bottom of the gradient (F4 + F5). Taken together, these results indicate that the TMs are largely responsible for producing the extreme curvature of LPPs generated at high protein concentrations. At lower Yop1 concentrations, the deletion of the APH leads to the conversion of tubules into larger vesicles.

Next, we performed similar experiments with *Xenopus* REEP5. Interestingly, this protein caused even higher membrane curvature than Yop1, generating mostly LPPs and a few larger vesicles, but

essentially no tubules (Fig. 3e). In a flotation experiment, the bottom fractions consisted almost entirely of LPPs (Fig. 3f; F4 + F5; flotation pattern see Fig. 3c). In the top fractions, larger vesicles were seen from which LPPs appear to bud off (Fig. 3f; F2). Thus, *Xenopus* REEP5 seems to be more efficient than Yop1 in generating LPPs. This conclusion was confirmed with reconstitution experiments using different protein to lipid ratios, in which the samples were analyzed by negative-stain EM and flotation (Supplementary Fig. 3a). *Xenopus* REEP5 generated LPPs at even a 1:4000 protein to lipid ratio (Supplementary Fig. 3a), whereas Yop1 required considerably higher concentrations (1:80; Supplementary Fig. 3a). *Xenopus* REEP5ΔAPH also generated LPPs at high concentrations (Supplementary Fig. 3b, c). Together, these results show that the REEPs can generate extreme curvature, but the efficiency depends on the species and concentration. For both Yop1 and *Xenopus* REEP5, the APH facilitates LPP formation but is not absolutely essential.

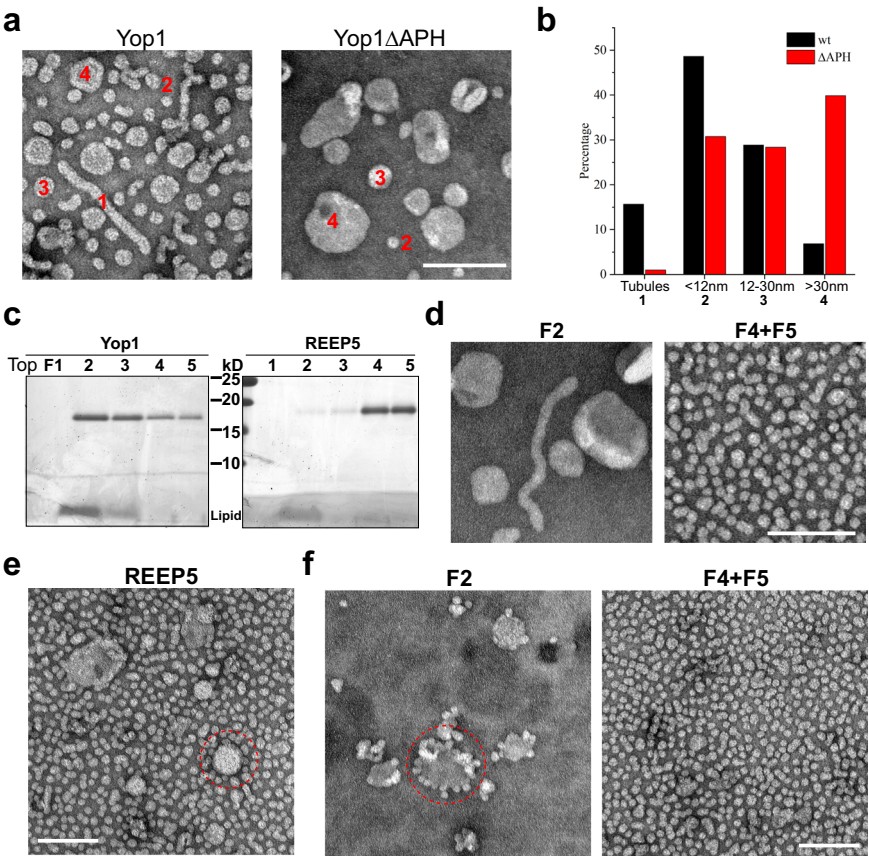

**Fig. 3 Reconstitution of purified REEPs generates high-curvature structures. a** Yop1 or Yop1ΔAPH were purified from DDM-solubilized membranes and reconstituted with *S. cerevisiae* lipids at a molar ratio of 1:200 protein:lipid. The samples were analyzed by negative-stain EM. Short tubules (1), vesicles with diameters <12 nm (2), vesicles with diameters 12–30 nm (3), and vesicles with diameter >30 nm (4) are highlighted. **b** Quantification of experiments as shown in **a** (*n* = 2664 for wt and 2632 for ΔAPH from three independent experiments). **c** Yop1 or *Xenopus* REEP5 were reconstituted with *S. cerevisiae* lipids at a molar ratio of 1:200 protein:lipid. The samples were subjected to flotation in a Nycodenz gradient, and fractions were analyzed by SDS-PAGE and Coomassie-blue staining. The band at the bottom is lipid. **d** The indicated fractions of the Nycodenz flotation of Yop1 in **c** were analyzed by negative-stain EM. **e** Negative-stain EM analysis of *Xenopus* REEP5 after reconstitution, but before flotation. **f** Fractions of the Nycodenz flotation of *Xenopus* REEP5 in **c** were analyzed by negative-stain EM. Dashed circles in **e** and **f** highlight LPPs budding off from large proteoliposomes. All experiments were performed three times. Bars, 100 nm.

**REEPs generate LPPs in cells**. *Xenopus* REEP5 could generate LPPs even when the protein was never exposed to detergent. When REEP5 was overexpressed in *E. coli* and the lysate subjected to centrifugation, a large fraction of the protein did not sediment with the membranes; up to 75% of the total population remained in the supernatant in LPPs (Fig. 4a, b). In contrast, Sey1 sedimented almost quantitatively (Fig. 4a, b), as expected for an integral membrane protein. Deletion of the APH from REEP5 reduced LPP formation but did not abolish it (Fig. 4a), as in the reconstituted system (Supplementary Fig. 3b, c). Full-length REEP5 could be purified from the supernatant fraction without detergent, using the same protocol as for the material solubilized in detergent. After affinity-purification on streptavidin beads and SEC, essentially only one protein band was seen in a Coomassie-blue stained SDS gel (Fig. 4c, d). The protein had an apparent molecular weight of ~400 kDa in SEC. In negative-stain EM, the peak fractions contained almost spherical particles of relatively homogeneous size (inset of Fig. 4c; diameter 7–12 nm). Extraction with chloroform/methanol followed by thin-layer chromatography (TLC) indicated that these particles also contain lipids, mostly phosphatidylethanolamine (PE) and phosphatidylglycerol (PG) (Fig. 4e), a composition similar to that of *E. coli* membranes. Based on the molar ratio of REEP5 to phospholipid molecules (1:30), and the size of the particles, we

estimate that each particle contains about 8 molecules of protein and 240 molecules of lipids. REEP5 in LPPs is fully functional, as it forms a GTP-dependent membrane network when co-reconstituted with the fusion GTPase Sey1[11] (Fig. 4f). As expected, no lipid bilayer could be detected when the LPPs were analyzed by cryo-EM. In contrast, a bilayer was obvious in even small liposomes (diameters <20 nm; Supplementary Fig. 4a). These data show that *Xenopus* REEP5 generates the extreme membrane curvature of LPPs even without solubilization and reconstitution; it is simply the high concentration of REEP5 that converts phospholipid bilayers into LPPs.

To further estimate the number of REEP molecules per LPP, we inserted the folded BRIL protein into the cytosolic loop between TM2 and TM3 of *Xenopus* REEP5. Overexpressed REEP5-BRIL generated LPPs as efficiently in *E. coli* as wild-type REEP5 (Supplementary Fig. 4b). Again, the particles had a size range of 7–12 nm in negative-stain EM. The addition of Fabs directed against the BRIL protein showed particles with a variable number of bound Fab molecules (peak at 4–6 molecules; examples are shown in boxes of Supplementary Fig. 4b). Considering that not all bound Fab fragments are visible in these EM images, this number is consistent with our other estimate of the number of REEP molecules per particle.

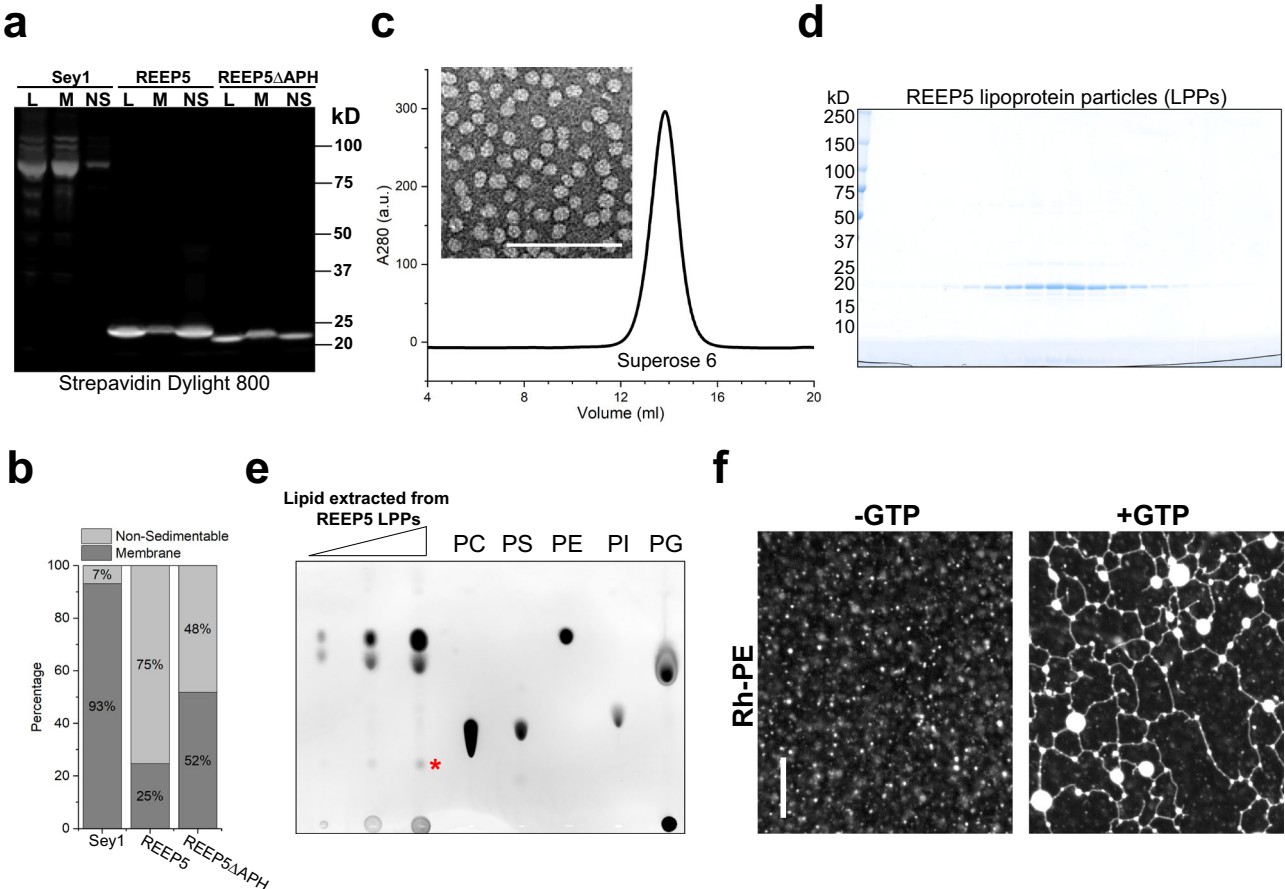

**Fig. 4 Overexpression of REEP/Yop1 in *E. coli* generates lipoprotein particles (LPPs). a** SBP-tagged REEP5, REEP5ΔAPH, or Sey1 were expressed in *E. coli*. Lysates (L) were fractionated by centrifugation into a non-sedimentable fraction (NS) and membrane fraction (M). Equivalent aliquots of these fractions were analyzed by SDS-PAGE, followed by blotting with Dylight 800-labeled streptavidin. **b** Quantification of the data in **a**. The experiment was performed two times. **c** *Xenopus* REEP5 was expressed in *E. coli* and purified from the NS fraction. It was then subjected to size-exclusion chromatography (SEC) and its elution followed by the absorbance at 280 nm. The inset shows a representative negative-stain image of the peak fraction. The visualized structures are referred to as lipoprotein particles (LPPs). Bar, 100 nm. **d** Fractions of the SEC run in **c** were analyzed by SDS-PAGE and Coomassie-blue staining. The experiment was performed four times. **e** Lipids were extracted from the purified LPPs, and different amounts of the extract were analyzed by thin-layer chromatography (TLC) and Primuline-staining. Pure PC, PS, PE, PI, and PG were run in parallel as markers. The asterisk indicates an unidentified lipid. The experiment was performed two times. **f** Purified LPPs were co-reconstituted with the GTPase Sey1 into liposomes containing POPC, DOPE, DOPS, and fluorescent rhodamine-PE (Rh-PE). The samples were incubated with or without GTP and visualized with a fluorescence microscope. The experiment was performed two times. Bar, 10 μm.

As with reconstituted proteins (Fig. 3), LPP formation in *E. coli* was less efficient with Yop1 than with *Xenopus* REEP5 (12% versus 75% remaining in the supernatant after centrifugation; Supplementary Fig. 4c). We also tested LPP formation in *E. coli* with REEP5 proteins from *T. thermophila* and *T. terrestris*. Again, species differences were observed, with REEP5 from *T. terrestris* being considerably more active in LPP formation, despite being expressed at a lower level (33% in the non-sedimentable fraction; Supplementary Fig. 4c). Negative-stain EM showed that the size of the LPPs formed by Yop1 and *T. ter* REEP5 was about the same as with *Xenopus* REEP5 (Supplementary Fig. 4d).

Interestingly, the interaction between the REEP monomers seems to be different in LPPs and membranes. We introduced a photoreactive probe into TM2 of *Xenopus* REEP5 or Yop1 and tested for dimer cross-links in the non-sedimentable LPP fraction and sedimentable membranes (Supplementary Fig. 4e). The non-sedimentable population reproducibly gave higher cross-linking yields, suggesting that most molecules are more tightly associated in LPPs than in low-curvature membranes. Taken together, these results indicate that REEPs generate LPPs when present at high concentrations, likely because REEP dimers generate so much

curvature that they break up phospholipid bilayers into mono-layers. The TMs seem to be the major factor for generating the high membrane curvature in LPPs, but the APH also contributes.

**The APH is required to generate a tubular membrane network.** To further analyze the role of the APH in curvature generation by the REEPs, we employed a reconstituted system in which the tubular ER network can be formed with liposomes containing purified proteins[11]. Full-length *Xenopus* REEP5 was reconstituted together with the fusion GTPase Sey1 into proteoliposomes containing a hydrophobic fluorescent dye. Addition of GTP resulted in the formation of a tubular membrane network that could be visualized in a fluorescence microscope (Fig. 5a). No network was seen in the absence of GTP (Fig. 5a). Similar results were obtained with Yop1 (Fig. 5b). *Xenopus* REEP5 lacking the APH domain did not form a network (Fig. 5a), consistent with the postulated role of APH in generating high membrane curvature at endogenous protein concentrations[11,14]. Mutations in the hydrophobic side of Yop1's APH also reduced (I145S) or abolished (I145K) network formation (Fig. 5b). The APH of Yop1

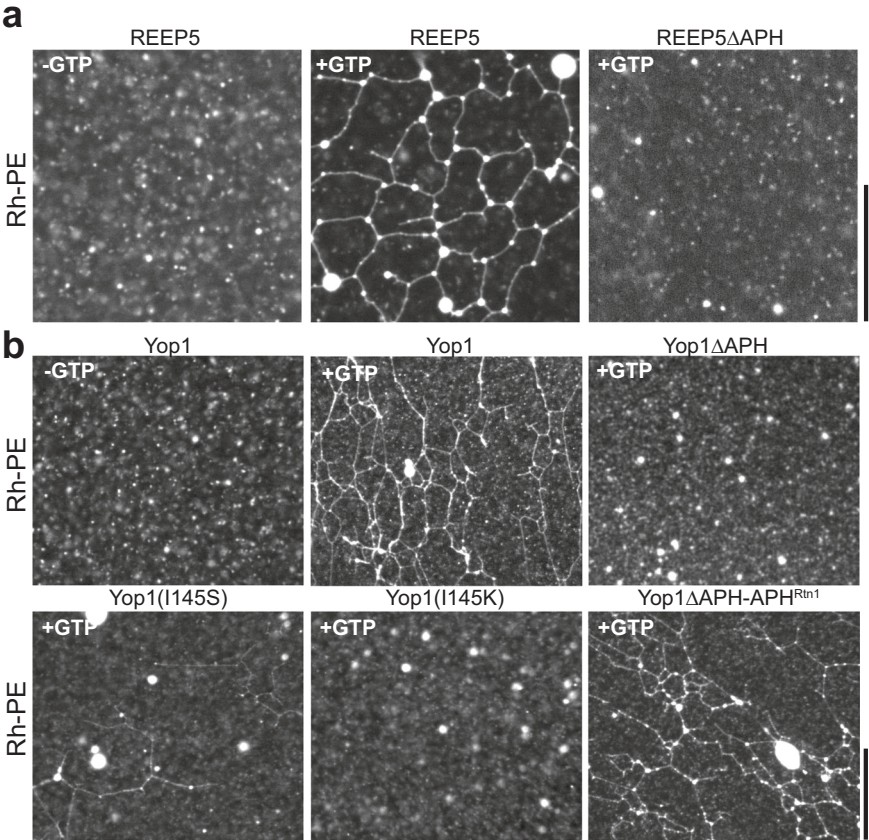

**Fig. 5 The APH of REEPs is required for the reconstitution of a tubular network. a** *Xenopus* REEP5 or REEP5 lacking the APH (REEP5ΔAPH) were purified from DDM-solubilized membranes and co-reconstituted with the GTPase Sey1 into liposomes containing POPC, DOPE, DOPS, and fluorescent Rh-PE. The samples were incubated with or without GTP, as indicated. The experiment was performed four times. **b** As in **a**, but with purified wild-type Yop1, a Yop1 mutant lacking the APH (Yop1ΔAPH), Yop1 variants with point mutations in the APH (Yop1(I145S) and Yop1(I145K)), or a Yop1 variant, in which the APH was replaced by that of sjRtn1 (Yop1ΔAPH-APH[Rtn1]). Bars (on the right), 10 μm. The experiment was performed two times.

could be replaced by that of *S. japonicus* Rtn1 (Fig. 5b), indicating that the APH plays a curvature-generating role in both tubule-shaping protein families.

The functional importance of the APH of REEPs was verified in vivo by analyzing the viability of *S. cerevisiae* cells. This organism was chosen because previous experiments had shown that the single or even combined deletion of the REEP and Rtn genes does not affect cell viability[5], but a triple knockout of Rtn1, Yop1, and the nucleoporin Nup85 is lethal[24]. Expression of wild-type *S. japonicus* Yop1 or *Xenopus* REEP5 from a centromeric plasmid rescued the viability of this triple mutant, whereas expression of mutants lacking the APH or harboring point mutations in the APH did not (Supplementary Fig. 5a). Similar results were obtained with *S. japonicus* Rtn1: wild-type Rtn1 rescued the viability of the triple mutant, whereas APH mutants did not (Supplementary Fig. 5a).

Further evidence for APH generating high membrane curvature was obtained with synthetic peptides. A peptide containing the wild-type APH sequence of Yop1 bound to liposomes, as demonstrated by its flotation in a Nycodenz gradient (Fig. 6a). Circular dichroism (CD) spectrometry indicated that the peptide formed a helix when bound to liposomes, but was unstructured in buffer or DDM (Fig. 6b). An APH peptide carrying the I145K mutation had a reduced propensity to form a helix (Fig. 6b), although it still floated with the liposomes (Fig. 6a). When wild-type peptide was added to liposomes containing reconstituted Yop1, and the samples were analyzed by negative-stain EM, the abundance of LPPs increased

and tubules and vesicles almost disappeared (Fig. 6c, d). In contrast, the I145K mutant peptide had little effect (Fig. 6c, d). Consistent with the increased number of LPPs, APH addition caused most of Yop1 to fractionate in heavy fractions of a Nycodenz gradient (Fig. 6e).

The APH can indeed contribute to the generation of excessive curvature that converts tubules into small membrane structures. This was demonstrated by adding a wild-type APH peptide of Yop1 to a reconstituted system containing both Yop1 and Sey1 (Fig. 7a). Instead of a membrane network, small vesicles and membrane fragments were observed that contained both lipid and protein (Fig. 7a and Supplementary Fig. 5b). In contrast, the I145K mutant peptide had no effect (Fig. 7a). A similar effect was seen when synthetic APH peptides were added to interphase *Xenopus* egg extracts that generate an ER network[12,25]. As in the reconstituted system, Yop1's wild-type APH peptide abolished the network (Fig. 7b–f), converting it into small membrane fragments (Fig. 7c). The I145K peptide was inactive (Fig. 7d). These results indicate that the amphipathic nature of the peptide is required to generate the high spontaneous curvature of the cytoplasmic membrane monolayer that breaks up the tubules into smaller membrane fragments.

A peptide corresponding to the APH of *Xenopus* REEP5 had the same effect as the Yop1 APH peptide, again converting the reticular ER network of interphase *Xenopus* egg extracts into small membrane fragments (Fig. 7e). To determine whether the APH promotes curvature by interaction with lipids or protein, we compared the effects of L- and D-amino acid APH peptides. We

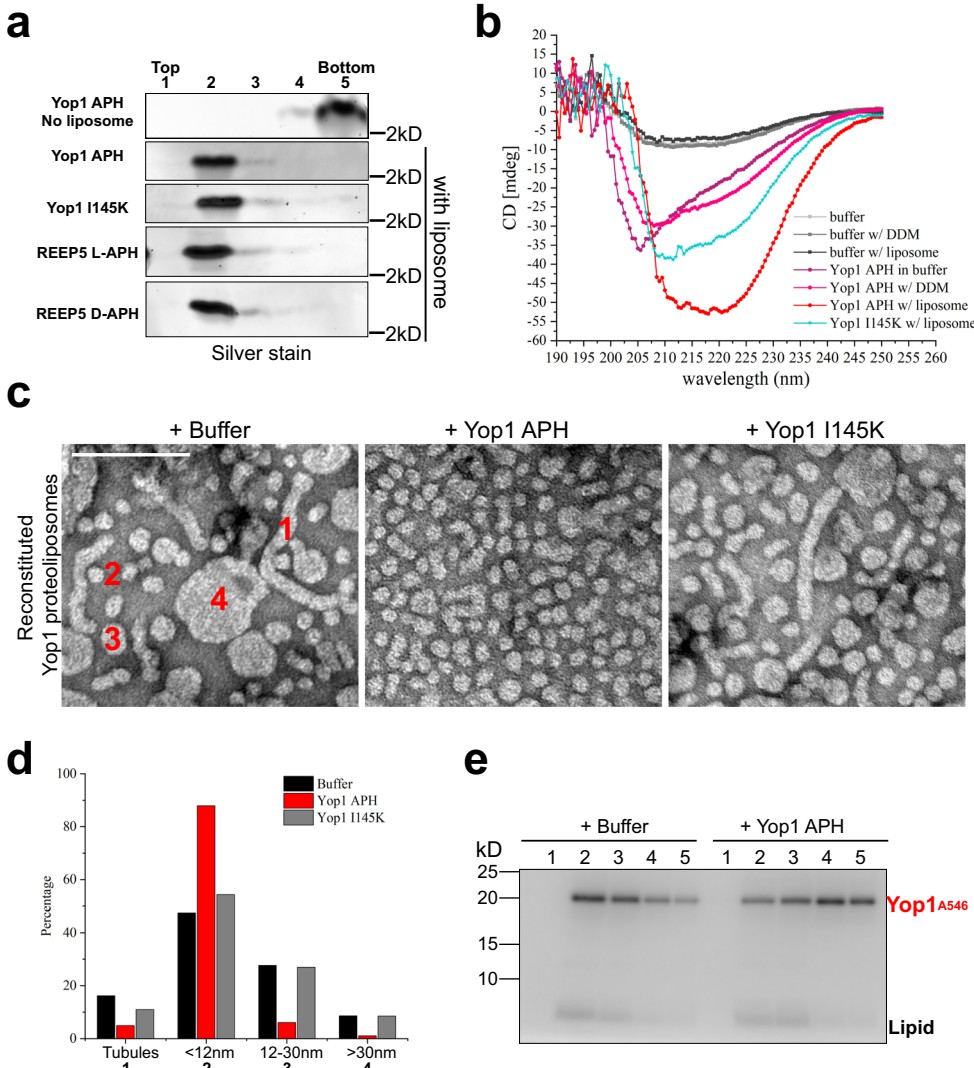

**Fig. 6 Synthetic APH peptides bind to liposomes and generate high curvature. a** Peptides corresponding to the APH of wild-type Yop1, to APH harboring a point mutation (Yop1 I145K), or to the APH of REEP5, with either L- or D-amino acids (REEP5 L-APH and REEP5 D-APH), were incubated with liposomes containing *S. cerevisiae* lipids at a 1:40 molar ratio of peptide to lipid. The samples were subjected to flotation in a Nycodenz gradient and analyzed by silver staining. The experiment was performed two times. The sequences of the peptides are shown in Supplementary Table 3. **b** The indicated peptides were analyzed by CD spectroscopy in the buffer, DDM, or after incubation with liposomes. The experiment was performed two times. **c** Yop1 was reconstituted into proteoliposomes and incubated with the indicated peptides. The samples were analyzed by negative-stain EM. Bar, 100 nm. **d** Quantification of the structures seen in **c**. Short tubules (1), vesicles with diameters <12 nm (2), vesicles with diameters 12–30 nm (3), and vesicles with diameter >30 nm (4) were counted ($n = 1662$ for a buffer, 1490 for Yop1 APH, and 1800 for Yop1 I145K from two independent experiments). **e** Alexa 546-labeled Yop1 (Yop1$^{A546}$) was reconstituted into liposomes containing *S. cerevisiae* lipids at a molar ratio of 1:200 protein to lipid. The samples were incubated with buffer or Yop1 peptide (1:20 molar ratio of peptide to lipid) and subjected to flotation in a Nycodenz gradient. Fractions were analyzed with a fluorescence scanner. The lower band corresponds to lipid. The experiment was performed three times.

found that a D-amino acid peptide of the same sequence as the L-amino acid peptide also disrupted the network (Fig. 7f). As expected, this peptide formed a helix that binds to liposomes (Fig. 6a), but it had the opposite chirality in CD spectrometry (Supplementary Fig. 5c). The L- and D-peptides abolished a network generated with metaphase *Xenopus* egg extracts, which normally contains both tubules and small sheets[12] (Fig. 7g–i). Given that both the L- and D-peptides convert tubules into structures of even higher membrane curvature, the APH likely acts by interaction with lipid, rather than protein. This is supported by experiments in which we added the L- and D-peptides to liposomes lacking proteins; negative-stain EM showed that large liposomes were converted into smaller vesicles or LPPs (Supplementary Fig. 5d).

Thus, at high concentrations, the APH alone can generate extreme curvature. Fractionation of the *Xenopus* extract, followed by immunoblotting for the endogenous REEP5 protein, showed that REEP5 is normally located entirely in sedimentable membranes, suggesting that LPPs do not form at physiological conditions. However, upon addition of the synthetic APH peptide, ~15% of the total population remained in the supernatant, likely in LPPs, indicating that the APH can generate extreme curvature, even when the curvature-generating proteins are present at endogenous concentrations (Supplementary Fig. 5e, f).

We also tested the APH of Pex11, which had been implicated in tubule formation of peroxisomes[26], in *Xenopus* extracts (Supplementary Fig. 5g). It acted similarly to the APH of the

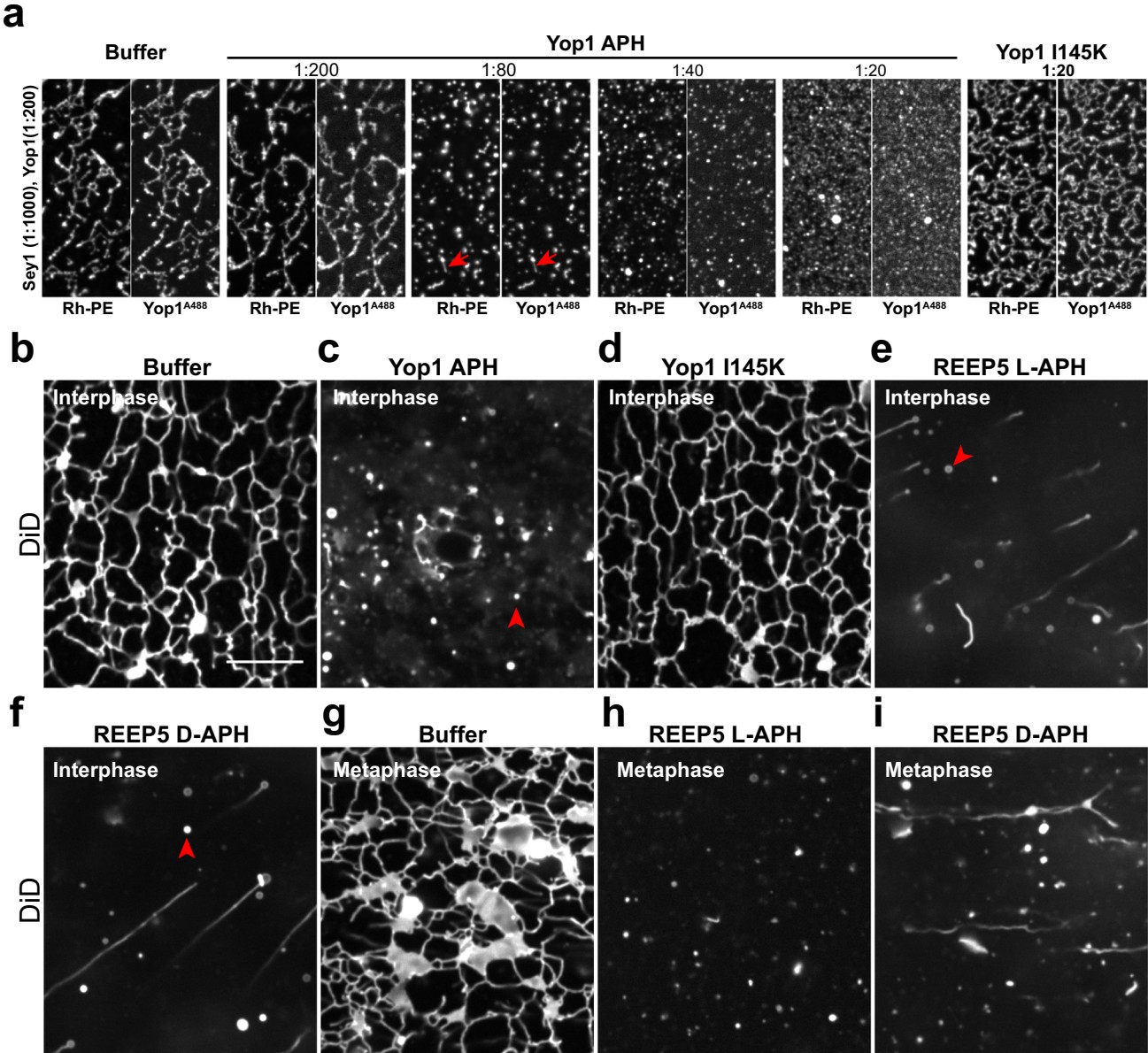

**Fig. 7 APH peptides abolish membrane networks in a reconstituted system and *Xenopus* egg extracts. a** Alexa 488-labeled Yop1 (Yop1$^{A488}$) was co-reconstituted with the GTPase Sey1 into liposomes containing *S. cerevisiae* lipids and fluorescent Rh-PE at the protein to lipid ratios given on the left. Synthetic peptides corresponding to the APH of wild-type Yop1 or to a mutant in the APH (Yop1 I145K) were added at the molar ratios (peptide to lipid) given on the top. All samples were incubated with Mg$^{2+}$ and GTP, added 10 min before the peptides. The samples were analyzed for fluorescence of lipid and protein. Arrows point to residual short tubules. The experiment was performed two times. **b** An interphase tubular network was generated with *Xenopus* egg extract and stained with the fluorescent dye DiD. The network was visualized with a fluorescence microscope with excitation at 642 nm. Bar, 10 µm. **c** As in **b**, but in the presence of 100 µM of a synthetic peptide corresponding to wild-type Yop1 (Yop1 APH). **d** As in **b**, but with a peptide carrying a point mutation in the APH (Yop1 I145K). **e** As in **b**, but with a peptide corresponding to the APH of wild-type *Xenopus* REEP5 (REEP5 L-APH). **f** As in **e**, but with a peptide containing D-amino acids (REEP5 D-APH). These experiments were performed four times. **g** Network generated with *Xenopus* egg extract in metaphase. **h** As in **g**, but in the presence of REEP5 L-APH. **i** As in **g**, but in the presence of REEP5 D-APH. These experiments were performed three times.

REEPs, again converting tubules into small membrane fragments. In contrast, a mutant peptide reported to be inactive in peroxisome tubulation (referred to as Pex11 3E because it carries Glu mutations at positions I69, I72, and F75[26]) was inactive (Supplementary Fig. 5g). Interestingly, the APH of Sey1 had no effect on the tubular network in *Xenopus* extracts (Supplementary Fig. 5g). In contrast to the APHs of the REEPs, this APH may not penetrate the membrane to an optimal depth to cause sufficient change in the monolayer spontaneous curvature[15].

**The APH is required for Rtn's exclusive ER-tubule localization and restricted mobility.** To test the role of the APH in vivo, we used the fission yeast *S. japonicus*, which has large cells that facilitate live-cell imaging[27]. *S. japonicus* has one member each of the Rtn and REEP protein families (Rtn1 and Yop1, respectively). We focused on Rtn1 because this protein localizes exclusively to the cortical tubular ER network and is absent from the nuclear envelope when expressed at endogenous levels (Fig. 8a); in contrast, Yop1 localizes to both the cortical ER and nuclear pores (Supplementary Fig. 6a, b), which complicates the analysis.

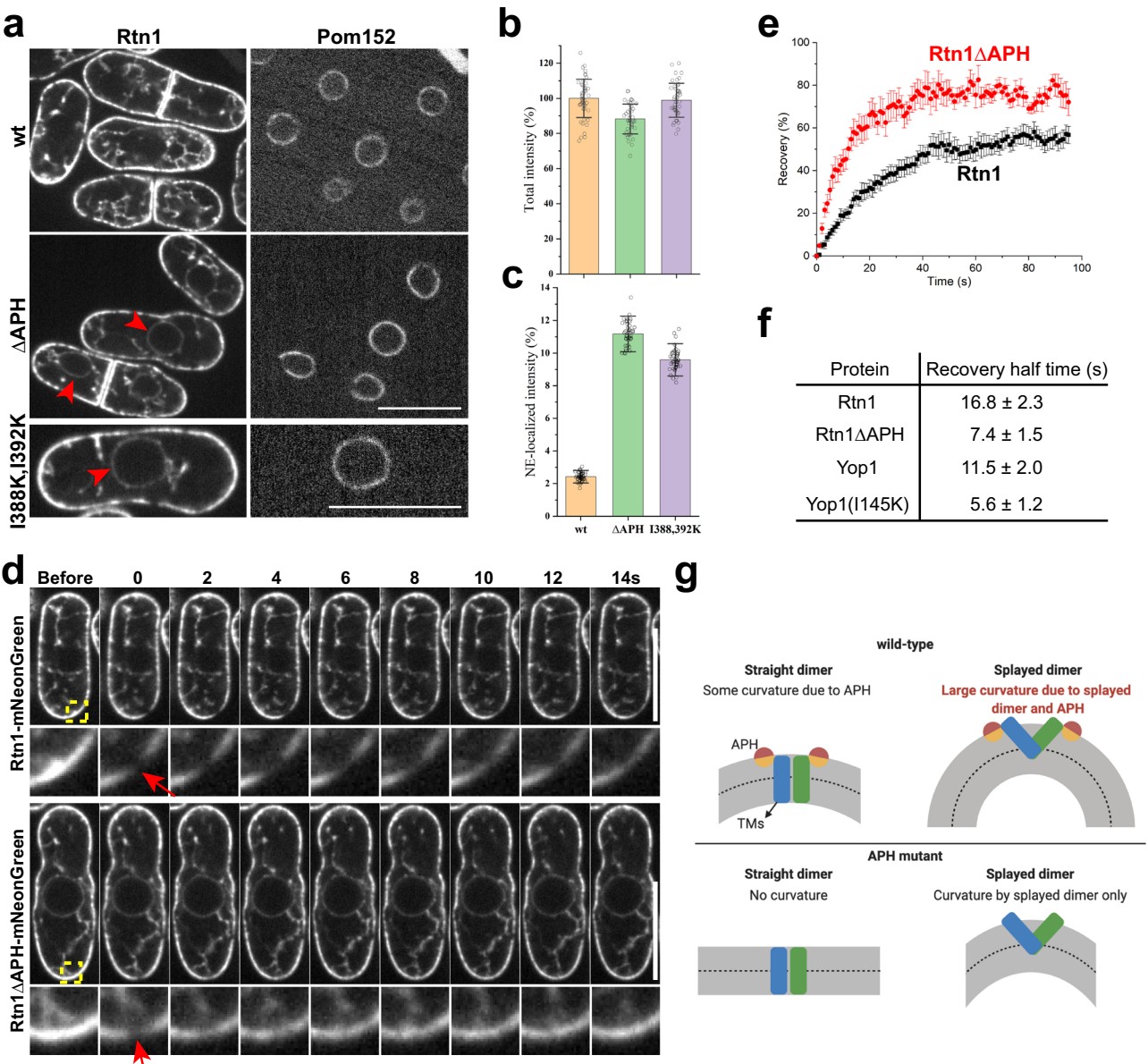

**Fig. 8 The APH is required for Rtn1's exclusive ER-tubule localization and restricted mobility in *S. japonicus* cells. a** Fusions of mNeonGreen with wild-type Rtn1 or APH mutants (ΔAPH, and I388K, I392K) were expressed in *S. japonicus* cells under *rtn1*'s native promoter. The cells also expressed a mCherry-fusion of Pom152 as a marker for nuclear pores and the nuclear envelope (NE). The cells were visualized with a confocal fluorescence microscope. Shown are representative images focused on the center of the cells obtained from three independent experiments. Arrowheads point to NE-localized mutant Rtn1. **b** Quantification of total fluorescence intensity. Background fluorescence acquired from untagged cells was subtracted. The total fluorescence of the mutants was normalized to that of wild-type cells. **c** Quantification of NE-localized fluorescence intensity. The Rtn1 fluorescence co-localizing with Pom152 was determined, the cytoplasmic background was subtracted, and the difference divided by the total intensity. Data are presented as mean ± SD of $n = 49$ (wt), 43 (ΔAPH), and 46 (I388K, I392K) cells for each strain from two independent experiments. **d** Cells expressing Rtn1-mNeonGreen or Rtn1ΔAPH-mNeonGreen were bleached at their periphery and the recovery of fluorescence followed over time (FRAP). The box shows the bleached area. The lower panels show magnified views. The arrows point to the bleached region of the peripheral ER. Bars, 10 μm. **e** Quantification of FRAP experiments. Fluorescence intensity before and immediately after bleaching was set to 100% and 0%, respectively. Error bars are standard error of the mean (SEM) of each time point from nine Rtn1-mNeonGreen-expressing cells and ten Rtn1ΔAPH-mNeonGreen-expressing cells. **f** Recovery half-times (± SEM) were calculated from FRAP experiments, such as shown in **d**. The analysis for Rtn1, Rtn1ΔAPH, Yop1, and Yop1 (I145K) is based on 9, 10, 14, 11 cells from three independent experiments, respectively. **g** Model for membrane curvature generation by REEP dimers. The TMs of REEPs can form either straight dimers (monomers represented by green and blue rods) that span the lipid bilayer (gray) or splayed dimers that sit predominantly in the outer leaflet of the bilayer. The APH (red/orange circle) promotes the formation of splayed dimers by generating local curvature through hydrophobic insertion. APH mutants can still generate membrane curvature when their local concentration is sufficiently high.

When the APH of Rtn1 was deleted, or when charges were introduced into the hydrophobic face of the helix, the protein was seen both in the cortical tubular ER network and the nuclear envelope (Fig. 8a; quantification in Fig. 8b, c). The Rtn1 mutant proteins localized to the low-curvature membranes of the nuclear envelope, rather than to the curved membranes around the nuclear pores (see scheme in Supplementary Fig. 6c): the deletion of the nucleoporin Nup132, caused pore clustering[28,29], but only small changes in the localization of the Rtn1 mutants (Supplementary Fig. 6d). These results, therefore, show that the APH is required for the exclusive localization of Rtn1 to the tubular ER, i.e., the region of high membrane curvature.

When the APH of Rtn1 was replaced by that of Yop1, the protein mostly localized to the cortical ER (Supplementary Fig. 6e; quantification in Supplementary Fig. 6f), indicating that it is partially functional. Similar effects were seen with the APH of Pex11, while the 3E mutant was inactive (Supplementary Fig. 6e, f). The amphipathic helix ALPS of the ArfGAP1 protein[30] was also inactive (Supplementary Fig. 6e, f); again, in contrast to the APHs of the reticulons and REEPs, this APH may not penetrate the membrane monolayer to an optimal depth.

Finally, we tested whether the APH affects the diffusional mobility of Rtn1 in the plane of the membrane. To this end, we expressed Rtn1 from the chromosome under its endogenous promoter as a fusion with the fluorescent protein mNeonGreen and performed Fluorescence Recovery After Photobleaching (FRAP) experiments on live cells. Upon bleaching of a small area, the recovery of fluorescence was followed over time (Fig. 8d). The recovery with a wild-type Rtn1 fusion was relatively slow, consistent with previous reports[14,21]. The recovery was considerably faster when the APH of Rtn1 was deleted (Fig. 8d; quantification in Fig. 8e, f). A fusion of Yop1 with mNeonGreen also diffused only slowly in the membrane, while the I145K mutant moved considerably faster (Fig. 8f). Thus, the APH seems to affect the mobility of both Rtn1 and Yop1, perhaps by promoting the formation of larger assemblies.

## Discussion

Here, we have analyzed the molecular mechanism by which the REEPs generate the high membrane curvature that is characteristic of ER tubules in cross-section. Our results show that the basic unit of the REEPs are dimers, rather than monomers, as previously believed. The two monomers interact within the membrane, with TM2 at the interface. Both features of the REEPs, the TMs, and the APH, contribute to generating high membrane curvature. The TMs alone can generate extreme curvature when present at high concentrations, converting phospholipid bilayers into micelle-like lipoprotein particles (LPPs). The APH is also capable of producing high membrane curvature, converting tubules into small vesicles or LPPs. As discussed below, these results have implications for how ER tubules are formed. Our conclusions are probably applicable to the Rtns as well because these proteins also have two hairpin TMs and an APH, they can replace the REEPs in forming an ER network, and Rtn1's APH is important for its function and can be swapped with that of Yop1. Nevertheless, it remains to be tested whether Rtns form stable dimers and can generate LPPs. The ER-phagy receptor FAM134B also has a similar structure as the REEPs and Rtns, and molecular dynamics simulations of a membrane-embedded monomer show that again both the TMs and the APH are required for curvature generation[31]. Although FAM134B formed clusters that locally bent the bilayer, no LPP formation was observed, perhaps because dimerization through the TMs was not considered.

The most surprising finding of our study is that, at high concentrations, the REEPs can generate so much curvature that a lipid bilayer is converted into a micelle-like lipoprotein particle (LPP). To our knowledge, the REEPs are the first transmembrane proteins that can form LPPs. NMR studies suggested that both TM hairpins span a phospholipid bilayer[13], but there are only a few hydrophilic or charged residues at the tips of the hairpins and the REEPs can therefore sit both in a bilayer with an aqueous lumen and a monolayer of a micelle lacking a lumen. Other known LPP-forming proteins, such as high-density lipoprotein, contain short 11 amino acid repeats that form amphipathic helices[32,33]. The apoprotein wraps around the lipid core, with the hydrophobic sides of the helices facing the lipids. A similar structure has been suggested for α-synuclein, which also has multiple amphipathic helical segments, but no TM domains[34,35]. α-synuclein can form LPPs of similar dimensions as those generated by the REEPs, and like the REEPs, can generate tubules and small vesicles from larger vesicles[34,35]. However, the cellular function of α-synuclein remains obscure. Our in vitro experiments suggest that the REEPs may form LPPs by budding from larger vesicles, a process that would be analogous to the formation of micelles by detergent addition to liposomes[36].

Previous models postulated that the REEPs and Rtns generate membrane tubules not only by hydrophobic insertions but also by forming arc-like scaffolds around tubules[6]. This model was largely based on the mechanism of BAR-domain proteins, which are cytosolic proteins forming crescent-shaped, rigid dimers that attach to the cytoplasmic face of a membrane and impose their curvature on that of the lipid bilayer[37,38]. This scaffolding mechanism requires that the bending rigidity of the oligomers is larger than that of the membrane and that the binding energy between the monomers is sufficiently strong. The REEP/Rtn dimers might form oligomeric assemblies, as they diffuse only slowly in the plane of the membrane and generate a ladder in cross-linking experiments[13,21]. Our FRAP experiments also suggest that the APH promotes the association of dimers, although the APH probably acts indirectly through altering local membrane stress, because it does not crosslink to any parts of the protein. However, even if the REEPs/Rtns formed oligomers, it seems unlikely that these would be rigid structures that could mold a lipid bilayer similarly to the BAR-domain proteins, as they lack folded cytosolic segments and would have to form a scaffold from the TM segments.

Although the TMs likely do not form a rigid scaffold, they play a major role in generating high membrane curvature, as Yop1 lacking its APH can still generate LPPs at high protein concentrations. The mechanism by which the TMs generate curvature likely relates to how they are accommodated in a lipid monolayer or bilayer. If the TMs were oriented perpendicular to the membrane surface, they would produce no or little cross-membrane asymmetry and thus no substantial membrane curvature (Fig. 8g; straight dimers). We, therefore, propose that a REEP dimer has the propensity to have a splayed conformation in which the two monomers, and therefore their TMs, are tilted relative to one another (Fig. 8g; splayed dimers). The splayed dimer would sit mostly in the cytoplasmic monolayer of the ER membrane and form a strongly asymmetric structure across the monolayer. Such a structure would generate monolayer spontaneous curvature that promotes bending of the cytoplasmic monolayer. Because of the trans-monolayer coupling in a lipid bilayer, the whole-membrane curves, while the luminal monolayer resists this bending. As a result, both monolayers accumulate elastic stress. At high protein concentrations, the spontaneous curvature of the cytoplasmic monolayer becomes so large that the resulting elastic stresses disrupt the bilayer, converting it into the highly curved monolayer of an LPP. This mechanism is analogous to that established for the curvature stress-induced conversion of bilayers into micelles during membrane solubilization with

common detergents[39,40]. A collective transition of non-splayed to splayed dimers would be facilitated by their dense packing, as there would be little lipids left that could resist membrane bending. Our results show that REEPs from different species can generate different spontaneous curvature, which may be explained by slight variations in the tilting of monomers within the dimer. Perhaps, this causes the diameter of ER tubules to differ among organisms. Our model of curvature generation by splayed dimers is different from the previously considered mechanisms of hydrophobic APH insertion and scaffolding.

One mechanism by which the APH generates high membrane curvature is hydrophobic insertion. Our experiments with synthetic peptides show that the addition of APH is sufficient to break up tubules or large liposomes into small vesicles or LPPs. Because APH peptides of different chirality have the same effect, the helix directly deforms the lipid bilayer, rather than acting indirectly on other parts of the protein. The APH likely shallowly inserts into the outer leaflet of the lipid bilayer, displacing the polar head groups of the surrounding phospholipid molecules, and causing this monolayer to locally acquire a large positive spontaneous curvature. The hydrophobic insertion mechanism of the APHs[15] has been proposed for REEPs and Rtns and was originally shown for epsin and N-BAR-domain containing proteins[37,41,42].

Our cross-linking results suggest that the APH also affects the relative orientation of the monomers in the dimer, suggesting that the APH not only acts through hydrophobic insertion but also by favoring a curvature-generating splayed dimer conformation. This is likely not a direct effect of the APH on the protein conformation, because no photo-cross-links were seen with any of the residues of the APH. Rather, the APH may affect the relative position and orientation of the REEP monomers indirectly by facilitating local membrane bending and thus promoting the splayed dimer conformation. In the absence of the APH, there would be a mixture of splayed and straight dimers. At high concentrations, APH-lacking molecules would still form splayed dimers through their TMs and thus generate LPPs (Fig. 8g). Interconversion between straight and splayed dimers is supported by the different cross-linking yields seen with REEPs localized to the monolayer of LPPs or the bilayer of membranes. Thus, we propose that the APH increases the local membrane monolayer spontaneous curvature both directly by hydrophobic insertion, and indirectly by promoting the transition of straight to splayed dimers.

The fusion GTPase ATL can also generate high membrane curvature, similarly to the REEPs and Rtns, as it alone is sufficient to form a GTP-dependent tubular membrane network in a reconstituted system[11]. Interestingly, ATL seems to have the same membrane topology as the REEPs and Rtns[43], and it forms dimers even in its GDP-bound state[44–46], likely by interactions of the TMs inside the membrane[47]. In the GDP-bound state, ATL causes the breakup of tubules into small vesicles or membrane fragments[12], indicating that its GTPase-dependent fusion activity counteracts the excessive curvature generated by the TMs. The APH of ATL also interacts with lipids, which stimulates fusion[47,48], but it does not disrupt the ER network in Xenopus extracts. Nevertheless, the similarities between ATL and the REEPs and Rtns suggests that all these proteins use splayed dimers to generate high membrane curvature.

In summary, we propose a model (Fig. 8g) by which REEPs generate high membrane curvature by two mechanisms, the formation of splayed dimers by the TMs, structures that are asymmetric across the membrane, and the hydrophobic insertion of the APH in the cytoplasmic leaflet of the bilayer. The splayed dimer and the APH each generate local monolayer spontaneous curvature, which leads to the local bending of the membrane. Ensembles of

such domains cause the formation of either cylindrical tubules or spherical vesicles. Under physiological conditions, tubules would be favored over vesicles. Tubules are favored kinetically, as the transition from an initially flat membrane into a cylindrical tubule does not require membrane fission, in contrast to the formation of a vesicle. Tubules could also be favored thermodynamically if the projection of the dimers onto the membrane plane has a non-circular shape. Such a lateral anisotropy would favor anisotropic over the isotropic realization of the spontaneous mean curvature. Tubules would also be favored over vesicles if the dimers interacted even weakly to form elongated assemblies. Factors promoting the breakup of tubules into vesicles include the rotational entropy of the dimers, which would counteract lateral anisotropy of the dimers, and a negative modulus of the Gaussian curvature predicted from the generation of positive spontaneous curvature in the cytoplasmic membrane leaflet, which would favor tubule fission into spheres[38]. Regardless of whether tubules are favored over vesicles kinetically or thermodynamically, our model predicts that the REEPs form large vesicles at low concentrations, tubules at intermediate concentrations, and small vesicles and eventually LPPs at high concentrations. At physiological concentrations, tubules would predominate. The proposed model is not only consistent with the in vitro data presented, but also with the fact that overexpression of an Rtn leads to long, narrow, and stiff tubules, which tend to break up into smaller fragments[12,21]. Our model does not require the dimers to form higher oligomers, but oligomerization would counteract entropic effects and could increase the curvature anisotropy, which would help to segregate REEPs and Rtns in tubules and sheet edges, despite them being able to diffuse into flat sheets.

## Methods

**Protein expression and purification (membrane fraction).** The codon-optimized gene for *S. japonicus* Yop1 was cloned into a modified pET21b vector that encoded a C-terminal 3C protease site followed by an SBP tag, as previously described[11]. A modified pET28b vector coding for a C-terminal TEV protease site followed by an SBP tag was used for the expression of *X. laevis* REEP5. Site-directed mutagenesis was used to generate mutants of Yop1 and REEP5. The sequence of BRIL was synthesized as a gene block and inserted at the desired position of Yop1 or REEP5 using Gibson assembly. Plasmids and primers used in this study are listed in Supplementary Tables 1 and 2.

Yop1 and REEP5 were expressed and purified as SBP fusion proteins in *E. coli* BL21-CodonPlus (DE3)-RIPL (Agilent). Cells were cultured in 2x YT media, and expression was induced at $OD_{600}$ ~0.6 with 250–400 μM isopropyl-β-D-thiogalactopyranoside (IPTG) at 16 °C for 16–18 h. The cell pellet was washed and sonicated in lysis buffer containing 25 mM HEPES/KOH pH 7.4, 150 mM NaCl, and protease inhibitors. The lysate was cleared with low-speed centrifugation and centrifuged at 44,000×g for 1 h to sediment membranes. The membranes were solubilized for 1 h with 1% n-dodecyl-β-maltoside (DDM). Insoluble material was removed by centrifugation at 50,000×g for 30 min. The clarified supernatant was incubated with streptavidin agarose resin for 1 h at 4 °C. The resin was washed with 10–15 volumes of lysis buffer containing 0.03% DDM. Protease inhibitors were left out for the last five washing steps. Proteins were eluted by on-column cleavage of SBP with 3C (Yop1) or TEV (REEP5) protease overnight at 4 °C in a buffer containing 25 mM HEPES/KOH pH 7.4, 100 mM NaCl, 0.03% DDM. In all experiments, 2 mM dithiothreitol (DTT) was added for the purification of *X. laevis* REEP5. Purification of *S. cerevisiae* Sey1 was as described in ref. [11]. All membrane proteins were further purified by SEC on a Superdex 200 or Superose 6 column (GE Healthcare) and concentrated by ultrafiltration (Amicon Ultra, EMD Millipore).

To incorporate Bpa probes into REEP/Yop1, the mutated REEP expression plasmid and the amber suppressor tRNA plasmid were co-expressed in BL21 (DE3) cells (NEB)[49]. The expression steps were similar to those with wild-type Yop1, except that 1 mM final concentration of H-Bpa-OH (Chempep) was added and amber suppressor was induced 1–2 h before IPTG addition.

Purified Yop1 in DDM was labeled with Alexa Fluor NHS ester dyes (Thermo Fischer Scientific) as described in ref. [11]. The labeling efficiency was calculated by comparing the absorbance of the protein at 280 nm and the absorbance of the dye at 556 nm for Alexa Fluor 546 and 495 nm for Alexa Fluor 488, using a NanoDrop 2000c Spectrophotometer (Thermo Scientific).

**Multiangle light scattering (MALS).** Affinity-purified REEP5 or Yop1 was subjected to gel filtration on a Superdex 200 column in a buffer containing 20 mM Tris/HCl pH 7.5, 100 mM NaCl, and 0.03% DDM. Approximately 100 μl fractions

of the peak were collected (protein concentration 0.2–1 mg/ml) and used for SEC-MALS. Protein samples were passed through an Agilent Advance Bio 300 column on an Agilent liquid chromatography system, a DAWN HELEOS II multiangle light scattering detector, an Optilab T-rEX refractive index detector, and UV detector (Wyatt Technology Corporation). Data were processed in ASTRA 6 using the protein conjugate model. We used d$n$/d$c$ values of 0.185 for proteins and 0.133 for DDM[50].

**Purification of Fabs directed to BRIL**. Generation and cloning of Fab fragments against the BRIL fusion protein were described in ref. [18]. Fab fragments were expressed in *E. coli* BL21-CodonPlus (DE3)-RIPL. Cells were cultured in 2x YT media at 37 °C, and the expression was induced with 1 mM IPTG for 4 h at 37 °C when the OD600 reached 0.6–0.8. Cells were lysed by sonication, and the lysate was cleared by centrifugation at 50,000×$g$ for 30 min. The supernatant was passed over 1–2 ml of Protein A Sepharose 4B resin (ThermoFisher), pre-equilibrated in 50 mM HEPES/KOH pH 7.5, 300 mM NaCl. The resin was washed with 10 column volumes of buffer. Fabs were eluted with 0.1 M acetic acid, and fractions were collected dropwise into 1.5-ml tubes containing buffer to bring the final concentration in each tube to 100 mM Tris/HCl pH 8.0, 100 mM NaCl. Fractions containing Fabs were combined, concentrated on Amicon 30 kDa molecular weight cut-off filters, and further purified by SEC on a Superdex 200 column equilibrated in 20 mM HEPES/KOH pH 7.5, 150 mM NaCl. All purification steps were carried out at 4 °C.

To confirm Yop1 is a dimer in detergent, SEC-purified Yop1-BRIL or Yop1 (I145R)-BRIL in DDM were incubated with a 2.5-fold molar excess of BRIL Fabs for 1 h at 4 °C, followed by subsequent chromatography on a Superose 6 column equilibrated in 25 mM HEPES/KOH pH 7.4, 100 mM NaCl, 0.03% DDM. The peak fractions were collected and diluted to 10 µg/ml for negative-stain EM analysis.

**Reconstitution of REEPs**. All lipids were obtained as chloroform stocks from Avanti Polar Lipids. Liposomes made of *S. cerevisiae* Yeast Polar Lipid Extract were prepared as previously described in ref. [11] with minor changes. The chloroform lipid mixtures were first dried under a nitrogen stream and then under vacuum overnight. Dried lipid films were hydrated in buffer containing 25 mM HEPES/KOH pH 7.4, 100 mM NaCl, 1 mM EDTA, and 5% glycerol and subjected to ten freeze–thaw cycles.

Detergent-mediated reconstitution was used to integrate all proteins into liposomes[11]. Briefly, liposomes and protein were incubated together at the desired protein:lipid ratio and supplemented with DDM to reach an estimated final concentration of ~0.1%. The mixture was incubated with gentle rotation at 4 °C for 1 h. The detergent was then removed by four successive additions of Bio-Beads SM-2 Resin (Bio-Rad) over the course of ~24 h at 4 °C or 4 h at room temperature. Insoluble aggregates were removed by centrifugation.

Nycodenz gradient centrifugations were performed as follows: 25 µl of the reconstituted sample or peptide-bound liposomes were mixed with 25 µl of 80% (w/v) Nycodenz and placed at the bottom of a Beckman polypropylene 5 × 20-mm tube. In total, 50 µl of 30%, 20%, 10%, and 0% Nycodenz were added sequentially on top. Samples were centrifuged at 50,000 rpm (~225,000×$g$) for 1 hr in Optima TLX ultracentrifuge using a TLS55 rotor. Five 50 µl fractions were collected from the top and analyzed by SDS-PAGE.

**Photo-cross-linking experiments**. Purified Bpa incorporated REEP/Yop1 mutants (5 µM) in buffer containing 0.03% DDM or reconstituted into liposomes generated with a *S. cerevisiae* lipid extract at a protein:lipid molar ratio of 1:200 were added to Thermowell PCR tubes (Corning) and placed on an ice-cold metal block. A long-wave UV lamp (Blak-Ray) was positioned 5 cm from the tubes, and the samples were irradiated for 30 min. After irradiation, SDS sample buffer was added. The samples were then subjected to SDS-PAGE and Coomassie-blue staining.

**Purification of REEP5 lipoprotein particles (LPPs) from *E. coli***. To purify *X. laevis* REEP5 LPPs from *E. coli*, the supernatant of the 44,000×$g$ centrifugation step was collected and incubated with streptavidin agarose resin with gentle rotation for 1 hr at 4 °C. The resin was washed extensively with 15 resin volumes of buffer containing 25 mM HEPES/KOH pH 7.4, 300 mM NaCl, 5 mM MgCl$_2$, 0.1% (w/v) ATP, 2 mM DTT, and protease inhibitors, then with 15 resin volumes of the buffer without protease inhibitors. LPPs were eluted by on-column cleavage of the SBP tag with TEV protease overnight at 4 °C. The eluted material was further purified by SEC on a Superose 6 column with lysis buffer supplemented with 2 mM DTT and 5% glycerol.

To determine the relative distribution of REEP5 in the sedimentable membrane and non-sedimentable supernatant fractions, SDS sample buffer was added to equivalent amounts of total lysate, membranes (resuspended in the same volume as the supernatant), or supernatant. The samples were heated at 65 °C for 10 min and then subjected to SDS-PAGE. Proteins were transferred to PVDF membranes with a Bio-Rad Turbo transfer system and probed with DyLight 800-labeled streptavidin (Invitrogen) diluted 1:2000 in 1% BSA. The membranes were scanned with an Odyssey imager (LI-COR).

To analyze the Fab-bound REEP5-BRIL LPP complex, SEC-purified LPPs were incubated with a tenfold molar excess of BRIL Fabs for 1 hr at 4 °C, followed by

chromatography on a Superose 6 column equilibrated in 20 mM HEPES/KOH pH 7.5, 150 mM NaCl. The peak fraction was collected and diluted to 5 µg/ml for negative-stain EM analysis.

**Lipid extraction and TLC**. Purified REEP5 LPPs containing 115 µg of protein was extracted with the Bligh and Dyer Lipid Extraction method[51], resulting in 50 µl of the organic phase. 1-, 4-, and 10 µl of the organic phase was loaded on a silica plate (EMD Millipore) and run with 60% chloroform/35% methanol/5% H$_2$O as a solvent. Lipids were stained with Primuline and visualized in a fluorescence scanner.

**Reconstitution of a tubular network with purified Sey1 and REEPs**. For reconstitution of a tubular membrane network, liposomes consisting of 65:18.5:15:1.5 mole percent of POPC:DOPE:DOPS: 1,2-dioleoyl-*sn*-glycero-3-phosphoethanolamine-N-[lissamine rhodamine B sulfonyl] (Rh-PE), or *S. cerevisiae* extract liposomes, consisting of 98.5:1.5 mole percent Yeast Polar Lipid Extract:Rh-PE were used as previously described[11]. These liposomes were extruded 21 times through polycarbonate filters of 100-nm pore size using a mini-extruder (Avanti Polar Lipids) at room temperature.

Co-reconstitution of Sey1 and REEPs was essentially done as described in ref. [11]. The proteoliposome suspension was first supplemented with 5 mM MgCl$_2$. Then, either 2 mM GTP or an equal volume of buffer was added, and the reaction was incubated for 5–10 min in the dark at room temperature. The reactions were mounted between two PEGylated coverslips, sealed with Valap, and placed on a microscope stage for another 5–10 min to allow settlement of the network prior to imaging.

Network formation with REEP5-containing LPPs was similar, except that purified REEP5 LPPs were co-reconstituted with Sey1 into Rh-PE-labeled POPC:DOPE:DOPS liposomes at a 1:50 molar ratio of REEP5 to lipid in liposomes, and that the total concentration of DDM was ~0.2% during the incubation.

**Fluorescence microscopy**. All fluorescence microscopy samples were visualized using a spinning-disk confocal mounted on a Ti-motorized inverted microscope (Nikon). Microscope settings were the same as previously described[11]. Green, red, and far-red fluorescence were excited with 488-nm, 561-nm, and 642-nm solid-state lasers. Images were acquired with a cooled CCD camera (Hamamatsu Photonics) controlled with MetaMorph and processed with Image J software.

**Peptide synthesis and circular dichroism (CD) spectroscopy**. Yop1 APH and I145K peptides were synthesized by the company Biosynthesis. Peptides corresponding to the APHs of *Xenopus* REEP5, Pex11, Pex11 3E, and of *S. japonicus* Sey1 were synthesized by the company Genscript. The sequences of the peptides are listed in Supplementary Table 3. The peptides were dissolved in a buffer containing 25 mM HEPES/KOH pH 7.4, 100 mM KCl at 2–4 mg/ml. The exact concentration of the stock solutions was determined by adsorption at 280 nm using a Nano Drop 2000c spectrophotometer.

For CD analysis, the Yop1 and REEP5 APH peptides were diluted to 0.2 mg/ml with buffer containing 25 mM Tris/HCl pH 8.0, 50 mM NaCl. Liposomes containing a final concentration of 0.6 mg/ml *S. cerevisiae* lipids or 0.3% DDM were added to test their effect on the secondary-structure formation of the peptides. The CD spectra were acquired on a Jasco J-815 spectropolarimeter equipped with a PFD-425S/15 Peltier unit. Spectra were accumulated 5 times from 250–200 nm, using a scanning rate of 50 nm/min. Measurements were performed at 20 °C in a 1-mm quartz cuvette.

**ER network formation with *Xenopus* egg extracts**. An interphase or metaphase ER network was formed using crude *Xenopus* egg extracts, as described previously[12,52,53]. The network was stained with DiD (1,1'-dioctadecyl-3,3,3',3'-tetramethylindodicarbocyanine, 4-chlorobenzenesulfonate salt). The samples were mounted on PEGylated coverslips and visualized by a spinning-disk confocal microscope at a far-red channel. To test the effect of synthetic peptides, a final concentration of 100 µM peptide was added to DiD-labeled interphase or metaphase extracts, and the samples incubated at room temperature for 5–10 min before imaging.

To determine the relative distribution of REEP5 in *Xenopus* egg extracts between a sedimentable membrane fraction and a non-sedimentable fraction, immunoblotting was performed with a polyclonal REEP5 antibody from Proteintech (used at 1:1000 dilution). Control blots were performed with calnexin antibodies (Enzo) (1:800 dilution) and with polyclonal antibodies raised against *Xenopus* Pex14 in rabbits, which were affinity-purified and used at a 1:10,000 dilution.

**Negative-stain EM and Cryo-EM**. For negative-stain EM analysis of reconstituted proteoliposomes or purified LPPs, 5 µl of the sample was added to a glow-discharged carbon-coated copper grid (Pelco, Ted Pella Inc.) for 1 min. The excess sample was blotted off with filter paper. The grids were washed once with deionized water, and then stained once with freshly prepared 1% uranyl acetate for 30 s prior to blotting and air-drying. To determine the effect of peptides on naked liposomes by negative-stain EM, samples were applied to

glow-discharged grids in a small petri dish with three drops of 2% osmium acid on the side without direct touching. Images were collected using a conventional transmission electron microscope Tecnai G² Spirit BioTWIN operated at an acceleration voltage of 80 kV.

For negative-stain EM analysis of Fab-bound REEP5-BRIL LPPs and Fab-bound Yop1-BRIL in DDM, 3.5 µl of the sample was applied to grids for 45 s. Excess sample was removed, and grids were immediately washed with deionized water, followed by two applications of freshly filtered 1.5% uranyl formate solution for a total of 45 s prior to blotting and air-drying. Grids were imaged by a ThermoScientific Tecnai T12 transmission electron microscope operating at 120 kV and equipped with a Gatan UltraScan 895 CCD camera. Images were processed with RELION 3.0[54].

For cryo-EM, sonicated *S. cerevisiae* extract liposomes (2 mM) and purified REEP5 LPPs (0.2 mg/ml) were vitrified using a Vitrobot and Quantifoil R1/2 Copper grids. The grid was imaged in a FEI Tecnai F20 microscope operating at 200 KV. A total of 94 image stacks were acquired using a Gatan K2 Summit detector with a total dose of 50 e-/Å2. Motion-corrected images are then processed and analyzed with RELION 3.0[54].

***S. japonicus* strain construction, cell culture, and imaging**. Transformation of *S. japonicus* cells was performed as described[55,56], with ~10 µg of purified DNA fragments per transformation. All transformants were verified by PCR. Supplementary Table 4 lists the *S. japonicus* strains used in this study. DNA fragments for gene targeting and gene deletion were designed as described[56] with minor changes. Briefly, ~500–1000 bp homologous sequences flanking the designated insertion sites were used for homologous recombination. cDNA sequences carrying point mutations or short truncations were introduced into *rtn1* or *yop1* knockout strains to generate tagged Rtn1 or Yop1 APH mutants. In most cases, genetic crossing was used to generate double or triple mutants. In the case of *rtn1* and *pom152*, the genetic loci were too close, so that a second round of transformation was needed.

The rich medium YE5S (yeast extract with five supplements) was used for *S. japonicus* cell culture. Cells were grown in YE5S for ~24 h at 30 °C before imaging, with proper dilutions to keep the culture in log phase. Live-cell imaging was performed in Bioptechs 0.17-mm Delta T TPG Dishes with a small piece of YE5S agar on top of the cells. A z-focal plane series was acquired at 0.2–0.4 µm intervals spanning 8 µm in most cases, but the middle focal plane is shown in the figures unless otherwise indicated.

The triple-knockout *S. cerevisiae* strain *rtn1:Kan yop1:Kan nup85:HIS3 his3 ura3 leu2* harboring plasmid pSW3354 to express Nup85 was obtained from the laboratory of Susan Wente[24]. Yop1, *Xenopus* REEP5, *S. japonicus* Rtn1, and mutants were sub-cloned into the centromeric plasmid pRS415 under the control of *S. cerevisiae* Yop1's native promoter. This plasmid was transformed into yeast following the protocol described in ref. [57]. The cells were cultured in a synthetic medium lacking leucine and uracil. Serial fivefold dilutions were used to test growth on 5-fluoroorotic acid (5FOA) plates with an OD$_{600}$ of 0.5 at the highest concentration. The final concentration of 5FOA was 1 mg/ml.

**Fluorescence recovery after photobleaching (FRAP)**. FRAP experiments with *S. japonicus* cells were conducted with a spinning-disk confocal microscope and a Photonic Instruments MicroPoint laser targeting system. Laser alignment was performed before each imaging session. Magnification, laser power, and detector gains were identical across all samples. The microscope was focused on the middle of the cells. Two to four cells were selected with a region of interest (ROI) of ~10 × 10 pixels for each FRAP experiment. Five pre-bleaching and 100 post-bleaching images were acquired with 1-s intervals. Photobleaching was performed at 100% laser power, and the selected region was bleached to >50% of the original signal.

Data analysis was essential as described[58]. Briefly, after subtracting the background and correcting for the photobleaching during image acquisition, the ROI intensity was normalized with the mean pre-bleach intensity set to 100% and the intensity right after bleaching set at 0%. Time 0 indicates the end time of bleaching. The equation $y = m_1 + m_2 \exp(-m_3 x)$ was used to fit fluorescence recovery with Origin ($m_3$ is the off-rate). The half-time of recovery was calculated as $t_{1/2} = \ln2/m_3$.

**Reporting summary**. Further information on research design is available in the Nature Research Reporting Summary linked to this article.

## Data availability
Data supporting the findings of this paper are available from the corresponding author upon reasonable request. A reporting summary for this article is available as a Supplementary Information file. Source data are provided with this link https://drive.google.com/file/d/1iyqh68ZPwDM2cHQv7_7y46QdgHhMvREK/view?usp=sharing.

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

## Acknowledgements

We thank Snezhana Oliferenko, Sophie Martin, I-Ju Lee, Jian-Qiu Wu, and Susan Wente for sharing yeast strains and reagents; the Nikon Imaging Center, the Center for Macromolecular Interactions, and the Electron Microscopy Facility at Harvard Medical School for technical assistance; and Bradley French, and Yoko Shibata for critical reading of the manuscript. L.D.C was supported by a fellowship from the American Heart Association (20POST35140000), Y.G. by a fellowship from the Damon Runyon Cancer Research Foundation (DRG-[2354-19]), M.M.K by the Israel Science Foundation (3292/19) and the Deutsche Forschungsgemeinschaft (DFG) (SFB 958), and T.S. by the Israel Science Foundation (2751/20). T.A.R. is a Howard Hughes Medical Institute investigator.

## Author contributions

N.W. designed and performed essentially all in vitro and yeast experiments; L.D.C. purified REEP fusions and performed EM experiments; Y.G. collected cryo-EM images; N.W., L.D.C., Y.G., and T.A.R. analyzed the data; M.M.K., T.S., and T.A.R. interpreted the data and wrote a draft of the paper. All authors contributed to the writing of the paper.

## Competing interests

The authors declare no competing interests.
