## [Peer Review File · Nature Communications]

Reviewer #1 (Remarks to the Author):

The manuscript describes experiments investigating the molecular basis of membrane curvature generation in the Rtn and REEP families, which are responsible for ER tubule formation. The topic is an interesting and one of fundamental importance in cell biology. The authors report a number of interesting insights, providing novel and significant insights especially into the separate but cooperative effects of the TMs and the amphipathic helices (APH).

Among the key insights are the demonstration that these proteins form stable dimers mediated by TM2, that both the TM domain and the APH contribute to membrane curving, and that the coordination between the TM and the APH is likely indirect via the lipid membrane. APHs show up on most if not all of the proteins in this family, and the experiments provide valuable new insights into their role: cellular membrane localisation and reduction in mobility possibly through facilitating higher order oligomerisation. Another quite important insight from this work is the clarification that the very small vesicles that these proteins can produce, reported previously by this group and others, are actually lipoprotein particles rather than bilayered vesicles. The manuscript is a large leap forward in our understanding of these proteins.

I have just a few comments on the manuscript, especially about the wedging model. The authors suggest that the wedging model for generating membrane curvature by these proteins is probably more accurate than the scaffolding model, and a splayed helices model is presented in Fig. 8g. The model suggests that there are little or no stable intramolecular contacts between helices in the splayed conformation and that there is a continuum of possible splay angles in the membrane. It is a compelling model based on the available data. At the same time it is difficult to imagine what is happening in the LPPs if they are an extension of this model. It seems unlikely that LPPs could be the result of a 180 degree 'splay' because of the mismatch of the helix lengths with a monolayered micelle. Are the formation of LPPs reversible?

I also note that there is a paper by Hummer and colleagues (Bhaskara et al. 2019, Nature Comms) in which they show via simulations that a flexible reticulon homology domain, lacking stable helix-helix packing, is sufficient to induce membrane curvature. This seems relevant to this paper.

Also a point of clarity: The 'cross-membrane' asymmetry induced by splaying makes sense but it is not clear what is meant by asymmetry in the plane of the membrane. Wouldn't dimerisation remove any asymmetry in the membrane plane?

Finally, a minor point: it would be useful to use "Yop1" alone for all references to the *S. cerevisiae* protein and "sjYop1" for the other. For example, it seems the "Yop1" labels above Fig. 3a,c should be "sjYop1"?

Reviewer #2 (Remarks to the Author):

Wang and co-authors present a very interesting work combining in vitro and in cellulo experiments on the molecular origin of the membrane shaping effect of REEP proteins and to a more modest extent of Reticulons (Rtn). These molecules are essential for keeping the tubular shape of the ER. They both contain 2 pairs of trans-membrane domains embedded in the membrane and an

amphipatic helix (APH) on the C-terminus. Most of the work is performed on REEP proteins from fission yeast *S. japonicus* (Yop1) and on mutants of it, but in some experiments, *Xenopus* and also REEPs from fungi have been studied.

It had been proposed previously, but not shown, that REEPs and Rtns could assemble in oligomers with curved arc shapes to stabilize ER tubular shape. A wedging role for the APH was also proposed to contribute to membrane bending. So far, the molecular principle of ER shaping and the respective role of the TM domains and of the APH were not clearly identified yet.

The authors first use different assays to show that REEP proteins form homodimers, and not larger oligomers. Dimerization takes place through the very conserved TM2 domain and APH are not strictly required. Dimerization was shown not only in yeast but also in *Xenopus* and 2 sorts of fungi. They next study the membrane shaping capability of the REEP proteins. With negative staining EM, they show that WT Yop1 at high concentration transforms liposomes in tubules, small vesicles and even in micelles/lipoprotein particles (LPPs). APHs help to form LPPs. Similar extremely curved LPP structures are also observed in *E. coli* when REEPs are overexpressed. This allowed to quantify the number of proteins and lipids per micelles and the nature of the lipids. This clearly demonstrate the extreme shaping capability of these proteins, since they are able to completely fractionate the ER tubules at high density or when overexpressed. Finally, they study the role of the APHs in the formation of the tubular ER network. They used a reconstituted system with REEP and Sey1 required for fusion, that some of the authors previously developed and showed that the APH is required to form the network. This is also true in vivo. When APHs are added to the tubular network, they completely destroy it in small membrane fragments. Finally, they use Rtn1 to demonstrate that APHs are necessary for its correct localization to the tubular ER.

This work brings new insights about the REEP proteins, showing that they essentially form tight dimers but no stable larger oligomeric structures. They also show that these proteins have a strong shaping potential due probably to their splayed shape and if overexpressed, they can fragment the ER network and even form micellar lipoprotein structures. ER shape thus seems to result from a subtle balance between the wedging properties of the spalyed dimers and of the APH. Nevertheless, I have some issues and questions that should be addressed before being published in Nature Communications.

1- Could the authors discuss the origin of the large distribution of vesicle diameters (Fig. 3b)?

2- What so special about TM2 as compared to the other TMs? What type of interaction could occur through the membrane leading to tight dimer formation?

If proteins interact tightly through TM2 only, as suggested by the cross-linking experiments, doesn't imply an asymmetric shape of the dimer? Could this be added in the discussion?

3- Last part (lines 310-331):

- what it brings is not clear since localization of the Rtn and ER shaping are in principle intrinsically entangled. APH expression has an effect on ER morphology as shown by the authors, thus even if it also affects its localization to tubular ER when some mutant versions of the protein are expressed, in normal condition they localize on the structures they form.

- Overexpression of REEPs has dramatical effects on the shape of the ER. What is the Rtn expression level in the last series of experiments? Does it influence the geometry/shape of the ER?

- Rtn1 localized only on the cortical tubular ER and not on the nuclear pores in contrast with Yop1. Does it suggest different dimer geometry for the 2 proteins, one with affinity for negative Gaussian curvature (Yop1), and the other not? When the hybrid Rtn1 with Yop1 APH is expressed, it does not

change the localization of the protein to the cortical ER, suggesting that localization of Yop1 to the nuclear pores is not only dependent on its specific APH. Is it so clear as claimed in the discussion that "conclusions are likely applicable to Rtns"? I think the authors must present evidences that Rtn1 forms dimers to support this claim, although it is not so clear that this will be the case.

4- "Modeling this effect" (Cf. discussion): there is no model per se in the paper, but rather a physical discussion based on published papers of some of the authors. I think that a more careful use of "model" would be appropriate in the context of this manuscript.

5- Diffusion (see paragraph 375-388): In the discussion, the authors claim that the very slow diffusion of the REEPs in the ER membrane is due to the formation of oligomers. Isn't it possible that these dimers spontaneously self-assemble like it was suggested for BAR domains? In addition, the density of proteins is high and crowding effects could account for a very reduced mobility.

6- Atlastins (Lines 438-445): it is very surprising that, if ATs have a similar structure with APH like REEPs and form dimers, they would induce membrane fusion (with GTP) instead of membrane fragmentation like REEPs. I think that the conclusion of the paragraph is more than speculative. I would rather like to see a discussion/hypothesis on the origin of this difference.

Reviewer #3 (Remarks to the Author):

The manuscript by Wang et al. claims to describe the molecular mechanism by which the REEPs generate the high membrane curvature typical of ER tubules and, I am assuming, also of sheet edges. Unfortunately, the work does not deliver on its promise.

I appreciate the fact that the authors have done a lot of work to show micelle formation. However, they used very high, non-physiological protein concentrations and that could engender effects that have nothing to do with the biological function of REEPs. The main trouble is that micelles are not physiologically relevant and the real interest here would be to determine the mechanism whereby REEPs act on lipid bilayers. That REEPs also form micelles is of little interest from a cell biology view point.

In my opinion, the observation that REEPs form micelles does not add anything to previous work, also from the same lab, demonstrating that REEPs induce the curvature necessary for tubule formation. Furthermore, the role of APH in tubule generation had already been rather nicely shown both for reticulons and REEPs (Brady et al. 2015; Breeze et al. 2016) without recourse to LPP formation. This manuscript adds confusion to this issue as well as the role of TMs because it tries to sort out the relative contribution using micelle formation, a physiologically irrelevant phenomenon, as a criterion to evaluate APH function but fails to do so as shown by the following quotes (comments within parentheses)

a) "TMs are largely responsible for generating the extremely high curvature of LPPs, while tubule formation also requires the APH"

b) "APH facilitates LPP formation but is not absolutely essential" (does APH facilitate LPP formation or make tubules?)

c) "The TMs seem to be the major factor for generating the high membrane curvature, but the APH also contributes" (is extreme curvature dependent on TMs or APH?)

d) "When wild-type APH peptide was added to liposomes containing reconstituted Yop1 and the samples were analyzed by negative-stain EM, tubules and vesicles were converted almost

quantitatively into LPPs" (again, does APH make tubules or LPPs?)

e) "the amphipathic nature of the peptide is required to generate the high spontaneous curvature of the cytoplasmic membrane monolayer that breaks up the tubules into smaller membrane fragments" (does APH make or break tubules? This statement is rather hard to figure out if APH is required for tubule formation as stated above)

f) When the APH of Rtn1 (there is here another protein switch for in vivo studies motivated by the observation that Yop1 did not localize as the authors would have liked it to) was deleted, or when charges were introduced into the hydrophobic face of the helix, the protein was seen both in the cortical ER and the nuclear envelope. The Rtn1 mutant proteins localized to the low-curvature membranes of the nuclear envelope, rather than to the curved membranes around the nuclear pores.... These results therefore show that the APH is required for localization of Rtn1 to regions of high membrane curvature (now the TMs allow protein localization to flatter membranes despite being "largely responsible for generating the extremely high curvature of LPPs").

The formation of LPP in cells should be studied in living cells if it is meant to underscore the importance of the underlying process not in a bacteria overexpression paradigm where cells are broken up resulting in an experimental situation not unlike an in vitro system.

From these considerations, it follows that the authors have not been able to distinguish what the TMs and APH functions are. This information would have been critical to describe the promised mechanism of curvature generation. Even though they propose a model by trying to fit and discuss their data, the result is rather unsatisfactory since it's mostly based on unsupported speculations. Unfortunately, I cannot recommend publication of this paper in Nature Communications. In order to publish this work, the authors should re-evaluate the experiments and their interpretation, essentially submit a new paper. They need to ask what LPP formation truly means (is it an artifact? Is it really linked to generation of extreme curvature by a protein? If so, this needs a demonstration. How does it correlate with protein function within a bilayer? Does it have physiological significance and, if they believe so, what is it?). The authors should define the specific role of TMs. Also, how important is the contribution of each domain to the function of the protein? Unless answers to these questions are provided, this work will not be publishable because it raises more questions than it answers.

As an additional methodological comment, I would like to point out that it is very odd that the bulk of in vitro in a study focusing on membrane shaping is based on simple negative staining EM, a technique known to distort membrane shapes. The authors will need to substantiate their claims on membrane deformations by quantitative cryo-EM-based measurements, negative staining techniques are largely insufficient.

Brady, J.P., Claridge, J.K., Smith, P.G., et al. (2015) A conserved amphipathic helix is required for membrane tubule formation by Yop1p. *Proceedings of the National Academy of Sciences of the United States of America*

Breeze, E., Dzimitrowicz, N., Kriechbaumer, V., et al. (2016) A C-terminal amphipathic helix is necessary for the in vivo tubule-shaping function of a plant reticulon. *Proceedings of the National Academy of Sciences of the United States of America*

REVIEWER COMMENTS

Reviewer #1 (Remarks to the Author):

The manuscript describes experiments investigating the molecular basis of membrane curvature generation in the Rtn and REEP families, which are responsible for ER tubule formation. The topic is an interesting and one of fundamental importance in cell biology. The authors report a number of interesting insights, providing novel and significant insights especially into the separate but cooperative effects of the TMs and the amphipathic helices (APH).

Among the key insights are the demonstration that these proteins form stable dimers mediated by TM2, that both the TM domain and the APH contribute to membrane curving, and that the coordination between the TM and the APH is likely indirect via the lipid membrane. APHs show up on most if not all of the proteins in this family, and the experiments provide valuable new insights into their role: cellular membrane localisation and reduction in mobility possibly through facilitating higher order oligomerisation. Another quite important insight from this work is the clarification that the very small vesicles that these proteins can produce, reported previously by this group and others, are actually lipoprotein particles rather than bilayered vesicles. The manuscript is a large leap forward in our understanding of these proteins.

We appreciate the flattering comments of the reviewer.

I have just a few comments on the manuscript, especially about the wedging model. The authors suggest that the wedging model for generating membrane curvature by these proteins is probably more accurate than the scaffolding model, and a splayed helices model is presented in Fig. 8g. The model suggests that there are little or no stable intramolecular contacts between helices in the splayed conformation and that there is a continuum of possible splay angles in the membrane. It is a compelling model based on the available data. At the same time it is difficult to imagine what is happening in the LPPs if they are an extension of this model. It seems unlikely that LPPs could be the result of a 180 degree 'splay' because of the mismatch of the helix lengths with a monolayered micelle. Are the formation of LPPs reversible?

A 180 degree splay is not required to accommodate the helices in LPPs, particularly because they are relatively short and have few hydrophilic residues in the connecting loops. A maximum angle of 120 degrees would be sufficient.

I also note that there is a paper by Hummer and colleagues (Bhaskara et al. 2019, Nature Comms) in which they show via simulations that a flexible reticulon homology domain, lacking stable helix-helix packing, is sufficient to induce membrane curvature. This seems relevant to this paper.

We apologize for not having cited the paper by Bhaskara et al. We now mention their work on FAM134B, a protein postulated to be an ER-phagy receptor (p. 14). Several of their conclusions are consistent with ours, such as that both the TMs and APH are important for curvature generation and that the TMs can be highly tilted. However, they performed the MD simulations with a monomer and did not observe LPP formation, likely because they did not take dimerization into account. However, in their experimental negative-stain images, they actually see small structures that could correspond to LPPs. It should also be noted that FAM134B likely plays no role in tubular ER formation, as it is of low abundance.

Also a point of clarity: The 'cross-membrane' asymmetry induced by splaying makes sense but it is not clear what is meant by asymmetry in the plane of the membrane. Wouldn't dimerisation remove any asymmetry in the membrane plane?

We have replaced the term "in-plane asymmetry" by "lateral anisotropy", which is easier to understand. Lateral anisotropy would be caused if the projection of the dimer into the membrane plane had a non-circular shape. Such a lateral anisotropy could determine how a given mean curvature is realized, i.e. whether tubules or small vesicles are favored. This is now better explained in the Discussion (p. 17).

Finally, a minor point: it would be useful to use "Yop1" alone for all references to the *S. cerevisiae* protein and "sjYop1" for the other. For example, it seems the "Yop1" labels above Fig. 3a,c should be "sjYop1"?

All of our experiments were performed with *S. japonicus* Yop1, even the experiments with the triple knock-out in *S. cerevisiae* cells. So, we decided to refer to this protein as Yop1; otherwise, the labels in the figures become rather long.

Reviewer #2 (Remarks to the Author):

Wang and co-authors present a very interesting work combining in vitro and in cellulo experiments on the molecular origin of the membrane shaping effect of REEP proteins and to a more modest extent of Reticulons (Rtn). These molecules are essential for keeping the tubular shape of the ER. They both contain 2 pairs of trans-membrane domains embedded in the membrane and an amphipatic helix (APH) on the C-terminus. Most of the work is performed on REEP proteins from fission yeast *S. japonicus* (Yop1) and on mutants of it, but in some experiments, *Xenopus* and also REEPs from fungi have been studied.

It had been proposed previously, but not shown, that REEPs and Rtns could assemble in oligomers with curved arc shapes to stabilize ER tubular shape. A wedging role for the APH was also proposed to contribute to membrane bending. So far, the molecular principle of ER shaping and the respective role of the TM domains and of the APH were not clearly identified yet.

The authors first use different assays to show that REEP proteins form homodimers, and not larger oligomers. Dimerization takes place through the very conserved TM2 domain and APH are not strictly required. Dimerization was shown not only in yeast but also in *Xenopus* and 2 sorts of fungi. They next study the membrane shaping capability of the REEP proteins. With negative staining EM, they show that WT Yop1 at high concentration transforms liposomes in tubules, small vesicles and even in micelles/lipoprotein particles (LPPs). APHs help to form LPPs. Similar extremely curved LPP structures are also observed in *E. coli* when REEPs are overexpressed. This allowed to quantify the number of proteins and lipids per micelles and the nature of the lipids. This clearly demonstrate the extreme shaping capability of these proteins, since they are able to completely fractionate the ER tubules at high density or when overexpressed. Finally, they study the role of the APHs in the formation of the tubular ER network. They used a reconstituted system with REEP and Sey1 required for fusion, that some of the authors previously developed and showed that the APH is required to form the network. This is also true in vivo. When APHs are added to the tubular network, they completely destroy it in small membrane fragments. Finally, they use Rtn1 to demonstrate that APHs are necessary for its correct localization to the tubular ER.

This work brings new insights about the REEP proteins, showing that they essentially form tight dimers but no stable larger oligomeric structures. They also show that these proteins have a strong shaping potential due probably to their splayed shape and if overexpressed, they can fragment the ER network and even form micellar lipoprotein structures. ER shape thus seems to result from a subtle balance between the wedging properties of the splayed dimers and of the APH.

Thank you for the overall very positive assessment of the paper.

Nevertheless, I have some issues and questions that should be addressed before being published in Nature Communications.

1- Could the authors discuss the origin of the large distribution of vesicle diameters (Fig. 3b)?

We now mention in the text (p. 6) that the Yop1 protein is inserted into preformed liposomes. The size heterogeneity likely arises from different protein densities in the various structures. How exactly these density differences come about is unclear. Perhaps, the Yop1/detergent micelles have different sizes and Yop1 molecules are inserted as a group.

2- What so special about TM2 as compared to the other TMs? What type of interaction could occur through the membrane leading to tight dimer formation?

If proteins interact tightly through TM2 only, as suggested by the cross-linking experiments, doesn't imply an asymmetric shape of the dimer? Could this be added in the discussion?

We now mention that TM2 is indeed special, as it contains several conserved hydrophilic residues and a conserved Pro residue. As suggested by the reviewer, we now mention these points in the text (p. 5). We do mention in the Discussion that the dimers may be anisotropic (p. 17).

3- Last part (lines 310-331):

- what it brings is not clear since localization of the Rtn and ER shaping are in principle intrinsically entangled. APH expression has an effect on ER morphology as shown by the authors, thus even if it also affects its localization to tubular ER when some mutant versions of the protein are expressed, in normal condition they localize on the structures they form.

It is true that the proteins always “localize to the structures they form”, but without curvature generation (deletion or mutation of the APH), Rtn1 localizes throughout the ER, whereas the wild-type localizes only to high-curvature tubules. We slightly changed the text to clarify that Rtn1 segregates into high-curvature tubules (p. 12).

- Overexpression of REEPs has dramatical effects on the shape of the ER. What is the Rtn expression level in the last series of experiments? Does it influence the geometry/shape of the ER?

Rtn1 was expressed from the chromosome under its endogenous promoter. We now mention this in the text (p. 13).

- Rtn1 localized only on the cortical tubular ER and not on the nuclear pores in contrast with Yop1. Does

it suggest different dimer geometry for the 2 proteins, one with affinity for negative Gaussian curvature (Yop1), and the other not? When the hybrid Rtn1 with Yop1 APH is expressed, it does not change the localization of the protein to the cortical ER, suggesting that localization of Yop1 to the nuclear pores is not only dependent on its specific APH. Is it so clear as claimed in the discussion that "conclusions are likely applicable to Rtns"? I think the authors must present evidences that Rtn1 forms dimers to support this claim, although it is not so clear that this will be the case.

We have added data showing that wild-type Rtn1, but not APH mutants, rescues the viability of a triple-deletion mutant in *S. cerevisiae* (Supplementary Fig. 5a). These data show that the APH is generally important for both the Rtn and REEP family proteins. Nevertheless, we agree with the reviewer that we should be more cautious about extending our results to the Rtns and have therefore rephrased the text (p. 14). It is indeed intriguing that Yop1, but not Rtn1, localizes to nuclear pores. This could be due to intrinsic properties of the two proteins or to affinity of Yop1 for nuclear pore proteins. This is an issue of future research.

4- "Modeling this effect" (Cf. discussion): there is no model per se in the paper, but rather a physical discussion based on published papers of some of the authors. I think that a more careful use of "model" would be appropriate in the context of this manuscript.

We now use the word "model" only to refer to the scheme in Fig. 8g (p. 17).

5- Diffusion (see paragraph 375-388): In the discussion, the authors claim that the very slow diffusion of the REEPs in the ER membrane is due to the formation of oligomers. Isn't it possible that these dimers spontaneously self-assemble like it was suggested for BAR domains? In addition, the density of proteins is high and crowding effects could account for a very reduced mobility.

We now use a more cautious wording (p. 13).

6- Atlastins (Lines 438-445): it is very surprising that, if ATLs have a similar structure with APH like REEPs and form dimers, they would induce membrane fusion (with GTP) instead of membrane fragmentation like REEPs. I think that the conclusion of the paragraph is more than speculative. I would rather like to see a discussion/hypothesis on the origin of this difference.

ATL in fact causes membrane fragmentation in the absence of GTP (Wang et al., eLife). So, its fusion activity seems to counteract the curvature-generating activity. We now mention this fact in the Discussion (p. 17).

Reviewer #3 (Remarks to the Author):

The manuscript by Wang et al. claims to describe the molecular mechanism by which the REEPs generate the high membrane curvature typical of ER tubules and, I am assuming, also of sheet edges. Unfortunately, the work does not deliver on its promise.

I appreciate the fact that the authors have done a lot of work to show micelle formation. However, they used very high, non-physiological protein concentrations and that could engender effects that have nothing to do with the biological function of REEPs. The main trouble is that micelles are not physiologically relevant and the real interest here would be to determine the mechanism whereby

REEPs act on lipid bilayers. That REEPs also form micelles is of little interest from a cell biology view point.

We disagree with the reviewer. First, the fact that REEPs form LPPs at high concentration teaches us how they generate curvature. In all systems studied so far, it has been shown experimentally and theoretically that the curvature of a membrane is well represented by a weighted average of the individual curvatures of the membrane components. If the concentration of one component dominates, the resulting observed curvature is expected to be close to the curvature of this component. This justifies our focus on LLPs, as these experiments tell us how REEP dimers cause curvature. Ultimately, this insight translates into a better understanding of the cell biology of membrane shaping. Second, the REEPs are the first trans-membrane proteins known to form LPPs, which should be of great interest to people working on membranes, as it is only poorly understood how a bilayer is transformed into a monolayer.

In my opinion, the observation that REEPs form micelles does not add anything to previous work, also from the same lab, demonstrating that REEPs induce the curvature necessary for tubule formation. Furthermore, the role of APH in tubule generation had already been rather nicely shown both for reticulons and REEPs (Brady et al. 2015; Breeze et al. 2016) without recourse to LPP formation. This manuscript adds confusion to this issue as well as the role of TMs because it tries to sort out the relative contribution using micelle formation, a physiologically irrelevant phenomenon, as a criterion to evaluate APH function but fails to do so as shown by the following quotes (comments within parentheses)

a) “TMs are largely responsible for generating the extremely high curvature of LPPs, while tubule formation also requires the APH”

We changed the text on p. 6-7. We made clear that the APH promotes tubule formation at low REEP concentrations. At high REEP concentrations, LPPs are formed, and the APH has a stimulatory effect.

We had cited both papers mentioned by the reviewer (Brady et al. 2015 and Breeze et al. 2016), but we now quote them at additional places.

b) “APH facilitates LPP formation but is not absolutely essential” (does APH facilitate LPP formation or make tubules?)

The reviewer seems to think that LPP and tubule formation are contradictory. This is not the case. Tubules, small vesicles, and LPPs can all be produced by the same mechanism, depending on the intrinsic properties of the proteins (presence of the TMs and/or APH) and their local concentration. Generation of tubules, vesicles of the same radius and, finally, LPPs are sequential stages of increasing curvature. Therefore, breaking tubules into vesicles or generating LLPs does not mean the protein is doing something qualitatively different from tubule generation. We have changed a few places in the Discussion to clarify this point.

c) “The TMs seem to be the major factor for generating the high membrane curvature, but the APH also contributes” (is extreme curvature dependent on TMs or APH?)

The reviewer is apparently unhappy with our conclusion that both the TMs and the APH contribute to the generation of high membrane curvature. We have done extensive experiments to prove this point and our conclusions agree with those of Bhaskara et al. on FAM134B (see response to reviewer 1).

d) “When wild-type APH peptide was added to liposomes containing reconstituted Yop1 and the samples were analyzed by negative-stain EM, tubules and vesicles were converted almost quantitatively into LPPs” (again, does APH make tubules or LPPs?)

APH can generate membrane curvature that manifests itself in the form of tubules, small vesicles or LPPs (Supplementary Fig. 5d). We now mention this in the text (p. 11, 13).

e) “the amphipathic nature of the peptide is required to generate the high spontaneous curvature of the cytoplasmic membrane monolayer that breaks up the tubules into smaller membrane fragments” (does APH make or break tubules? This statement is rather hard to figure out if APH is required for tubule formation as stated above)

Again, the reviewer does not appreciate that tubules, small vesicles, and LPPs can all be produced by the same mechanisms, depending on the concentration of the REEPs in the membrane.

f) When the APH of Rtn1 (there is here another protein switch for in vivo studies motivated by the observation that Yop1 did not localize as the authors would have liked it to) was deleted, or when charges were introduced into the hydrophobic face of the helix, the protein was seen both in the cortical ER and the nuclear envelope. The Rtn1 mutant proteins localized to the low-curvature membranes of the nuclear envelope, rather than to the curved membranes around the nuclear pores.... These results therefore show that the APH is required for localization of Rtn1 to regions of high membrane curvature (now the TMs allow protein localization to flatter membranes despite being “largely responsible for generating the extremely high curvature of LPPs”).

The difference is simply due to the concentration of the Rtn1. LPP formation occurs at high protein concentrations, but the quoted experiments were done with Rtn1 expressed at endogenous levels.

The formation of LPP in cells should be studied in living cells if it is meant to underscore the importance of the underlying process not in a bacteria overexpression paradigm where cells are broken up resulting in an experimental situation not unlike an in vitro system.

We show that LPPs do not form in *Xenopus* extracts when the protein is expressed at endogenous levels. We do not claim that LPPs form under physiological conditions, and now mention this explicitly (p. 11-12). Nevertheless, LPP generation at high concentrations reveals the mechanism of curvature generation.

From these considerations, it follows that the authors have not been able to distinguish what the TMs and APH functions are. This information would have been critical to describe the promised mechanism of curvature generation. Even though they propose a model by trying to fit and discuss their data, the result is rather unsatisfactory since it's mostly based on unsupported speculations.

We provide evidence that the TMs cause the formation of splayed dimers and that the APH causes curvature both by insertion into the lipid bilayer and by promoting the conversion of straight into splayed dimers. So, our model scheme is based on data and is not based on “unsupported speculations”.

Unfortunately, I cannot recommend publication of this paper in Nature Communications. In order to publish this work, the authors should re-evaluate the experiments and their interpretation, essentially submit a new paper. They need to ask what LPP formation truly means (is it an artifact? Is it really linked to generation of extreme curvature by a protein? If so, this needs a demonstration. How does it correlate with protein function within a bilayer? Does it have physiological significance and, if they believe so, what is it?). The authors should define the specific role of TMs. Also, how important is the contribution of each domain to the function of the protein? Unless answers to these questions are provided, this work will not be publishable because it raises more questions than it answers.

These questions repeat the points raised before by the reviewer (see our response above). We strongly feel that the criticism is unjustified, as is also born out by the enthusiastic reports by the other two reviewers. The paper makes several new points:

1. LPP formation by the REEPs is a novel phenomenon that demonstrates that these proteins can generate extreme membrane curvature. We show that the TMs mediate LPP formation by forming splayed dimers.
2. The TMs and APH cooperate in generating high membrane curvature. The APH causes curvature by insertion into the lipid bilayer and by promoting the conversion of straight into splayed dimers.
3. Our results lead to a new model of curvature generation by the REEPs. This model is based on splayed dimers, rather than on a rigid scaffold formed by cytosolic domains.

As an additional methodological comment, I would like to point out that it is very odd that the bulk of in vitro in a study focusing on membrane shaping is based on simple negative staining EM, a technique known to distort membrane shapes. The authors will need to substantiate their claims on membrane deformations by quantitative cryo-EM-based measurements, negative staining techniques are largely insufficient.

We actually performed cryo-EM analysis of proteoliposomes containing reconstituted Yop1 or REEP5 and do provide cryo-EM images of LPPs and small liposomes generated by REEP5 (Supplementary Fig. 4a). These images show that negative-staining does not grossly distort the membrane structures. However, it is difficult to use cryo-EM to determine the number of the different structures and relate them to the protein concentration or their structural features, not only because of the extensive work that would be required. One fundamental issue is that the ice thickness affects the distribution of different membrane structures to a different extent. We found that tubules and large liposomes did not move into the holes of the EM grid at all, whereas the LPPs were more evenly distributed. Such a heterogeneous distribution is a common observation in cryo-EM and is caused by air interaction of the particles. It is thus not possible to derive a true distribution of the different structures from cryo-EM images. Another problem is that our reconstitutions lead to structures that likely contain different protein densities (see Figs. 3b and 6d) (see also response to reviewer 2). This means that quantification would be a convolution of structures of different densities and the size distribution at a given protein density. Finally, we would like to point out that negative-staining EM is the common technique used to study curvature-generating proteins, such as the BAR proteins. For example, the cited references 37-40 all use negative-stain EM. Negative stain EM was also used in Forst et al. (2008) 132, 807, Gallop et al. (2006) EMBO J. 25, 2898, Ferguson et al. (2009) Dev. Cell 17, 811, and in one of the papers pointed out

by the reviewer (Brady et al. 2015).

Brady, J.P., Claridge, J.K., Smith, P.G., et al. (2015) A conserved amphipathic helix is required for membrane tubule formation by Yop1p. *Proceedings of the National Academy of Sciences of the United States of America*

Breeze, E., Dzimitrowicz, N., Kriechbaumer, V., et al. (2016) A C-terminal amphipathic helix is necessary for the in vivo tubule-shaping function of a plant reticulon. *Proceedings of the National Academy of Sciences of the United States of America*

Reviewer #2 (Remarks to the Author):

I am very satisfied by the revised version of the paper.

Reviewer #3 (Remarks to the Author):

the main problem with the manuscript is that the authors claims are based on the assumption that formation of LPPs signifies high curvature generation. Unlike membranes, micelles don't have a bilayer and are normally generated in ways that do not involve curvature of a lipid bilayer. The authors need to demonstrate the tenet that extremely high curvature of a lipid bilayer results in the formation of single layer LPPs. If they provide such evidence, I will be happy to endorse the paper. In the absence of such evidence, I am unable to recommend publication.

Reviewer #3 (Remarks to the Author):

the main problem with the manuscript is that the authors claims are based on the assumption that formation of LPPs signifies high curvature generation. Unlike membranes, micelles don't have a bilayer and are normally generated in ways that do not involve curvature of a lipid bilayer. The authors need to demonstrate the tenet that extremely high curvature of a lipid bilayer results in the formation of single layer LPPs. If they provide such evidence, I will be happy to endorse the paper. In the absence of such evidence, I am unable to recommend publication.

The comment of the reviewer indicated to us that we had not sufficiently explained how high membrane curvature can convert lipid bilayers into LPPs. The mechanism outlined in the Discussion is not just speculation; it is based on our observation of LPP formation at high REEP concentrations, and on established principles of curvature-stress induced conversion of bilayers into micelles. Basically, if the TMs of the REEPs form a strongly asymmetric structure across the monolayer (splayed dimer), this would generate monolayer spontaneous curvature that promotes bending of the cytoplasmic monolayer. Because of the trans-monolayer coupling in a lipid bilayer, the entire membrane curves, while the luminal monolayer resists this bending. As a result, both monolayers accumulate elastic stress. At high protein concentrations, the spontaneous curvature of the cytoplasmic monolayer becomes so large that the resulting elastic stresses disrupt the bilayer, converting it into the highly curved monolayer of a LPP. This mechanism has been established for the curvature stress-induced conversion of bilayers into micelles during membrane solubilization with common detergents.

We added three sentences to the Discussion (p. 14) to better explain these points. We also added two references in which the mechanism is described:

39. Andelman, D., Kozlov, M. M. & Helfrich, W. Phase transitions between vesicles and micelles driven by competing curvatures. *EPL* **25**, 231–236 (1994).
40. Lichtenberg, D., Ahyayauch, H., Alonso, A. & Goñi, F. M. Detergent solubilization of lipid bilayers: A balance of driving forces. *Trends in Biochemical Sciences* 85–93 (2013).